# Microscale pH variations during drying of soils and desert biocrusts affect HONO and NH$_3$ emissions

Minsu Kim [1,2] & Dani Or [1]

Microscale interactions in soil may give rise to highly localised conditions that disproportionally affect soil nitrogen transformations. We report mechanistic modelling of coupled biotic and abiotic processes during drying of soil surfaces and biocrusts. The model links localised microbial activity with pH variations within thin aqueous films that jointly enhance emissions of nitrous acid (HONO) and ammonia (NH$_3$) during soil drying well above what would be predicted from mean hydration conditions and bulk soil pH. We compared model predictions with case studies in which reactive nitrogen gaseous fluxes from drying biocrusts were measured. Soil and biocrust drying rates affect HONO and NH$_3$ emission dynamics. Additionally, we predict strong effects of atmospheric NH$_3$ levels on reactive nitrogen gas losses. Laboratory measurements confirm the onset of microscale pH localisation and highlight the critical role of micro-environments in the resulting biogeochemical fluxes from terrestrial ecosystems.

[1] Soil and Terrestrial Environmental Physics (STEP), Department of Environmental Systems Sciences (USYS), ETH Zürich, 8092 Zürich, Switzerland. [2] Laboratory for Air Pollution/Environmental Technology, Empa (Swiss Federal Laboratories for Materials Science and Technology), 8600 Dübendorf, Switzerland. Correspondence and requests for materials should be addressed to M.K. (email: minsu.kim@empa.ch)

Nitrogen (N) is the most abundant element of Earth's atmosphere but occurs in an inert form (dinitrogen $N_2$) largely unavailable for common biological activity. $N_2$ gas is transformed into more reactive compounds (e.g., ammonium $NH_4^+$ and nitrate $NO_3^-$, collectively termed reactive nitrogen, $N_r$), that enable metabolism and growth of organisms. The transformation of $N_2$–$N_r$ occurs naturally in soils and is mediated by microorganisms. This crucial part of the nitrogen cycle entails nitrogen fixation, nitrification, and denitrification that produce various oxidation states of $N_r$[1,2]. The partitioning of N affects soil microbial communities and depends on environmental factors, such as soil type, organic carbon content, hydration, temperature, and pH[3,4]. Biologically fixed or imported $N_r$ in soils can be lost back to the atmosphere or leached to the ground by infiltrating water depending on the soil's environmental conditions. The soil nitrogen balance is important not only for soil fertility but also due to its roles in potent greenhouse gas emissions (e.g., nitrous oxide $N_2O$) and local pollutant dynamics to surface and groundwater resources (e.g., $NO_3^-$). Despite a vast body of research and observations, basic aspects concerning the fate of $N_r$ in soils and its environmental controls remain uncertain due to the complex interplay between biotic and abiotic processes.

The quantification of interactions between biotic and abiotic processes in soil remains a challenge, yet progress has been made in certain microbial systems, such as soil aggregates[5] and biological soil crusts[6] that help disentangle their role in ecosystem functioning. Biological soil crusts (hereafter biocrusts) have been suggested as a model microbial system to study microbial interactions at the community level within a well-defined domain (crust) under various abiotic conditions[7,8]. Biocrusts develop in cold and warm desert environments. Despite water limitations, these thin crusts host dense microbial communities and contribute significantly to biological $N_r$ exchanges with the atmosphere[9,10]. Considering that biocrusts are active only when wet, the partitioning and fate of imported $N_r$ during wetting events are of particular importance for their surrounding environments. N can be a limiting nutrient for desert ecosystems owing to relatively high loss of $N_r$ as gaseous emissions[11]. However, the picture of the nitrogen balance in biocrusts is more complicated due to strong effects of surface wetness, temperature, and community composition on $N_r$ dynamics[12].

N gaseous losses from desert soils include $N_2O$[13,14], nitric oxide (NO)[15–18], nitrous acid (HONO)[18,19], and ammonia ($NH_3$)[16,17,20]. $NH_3$ volatilisation has been shown to be the major loss of $N_r$ in deserts[16] owing to their high alkalinity (average pH ~ 8). Interestingly, biocrusts (largely alkaline with pH 6.8–8.2[18]) that mainly provide fixed N to desert soils also emit large amounts of HONO[18,19], known to form under acidic conditions because acid–base dissociation constant of HONO is pKa = 3.3[21,22]. This puzzle motivated our investigation of gaseous emission mechanisms regarding these two important soil nitrogen compounds, $NH_3$ and HONO. These are tightly coupled by (biotic) nitrification (the sequential oxidation of $NH_4^+$ to $NO_3^-$ with nitrite ($NO_2^-$) as intermediate product), and their pH dependency on degassing (abiotic protonation of $NH_3$ and $NO_2^-$) (Fig. 1).

Evidence suggests that significant amount of HONO can be emitted from neutral or alkaline soils (above pH ~ 5)[23]. This implies that general processes cause emissions of $NH_3$ and HONO from soils not limited to desert biocrusts. Moreover, the emission patterns of HONO during soil drying exhibit similar characteristics for different soil types, showing a peak emission at a certain optimal water content under unsaturated conditions. Studies have suggested that ammonia oxidisers are responsible for such distinct emission patterns of HONO from soils[23]. Scharko et al.[24] combined flux chamber measurements with genomic approaches and concluded that HONO emissions were related to the abundance of ammonia oxidisers within neutral or alkaline soils. Yet, their genomic analysis also indicated presence of active nitrite oxidisers that are expected to complete the nitrification

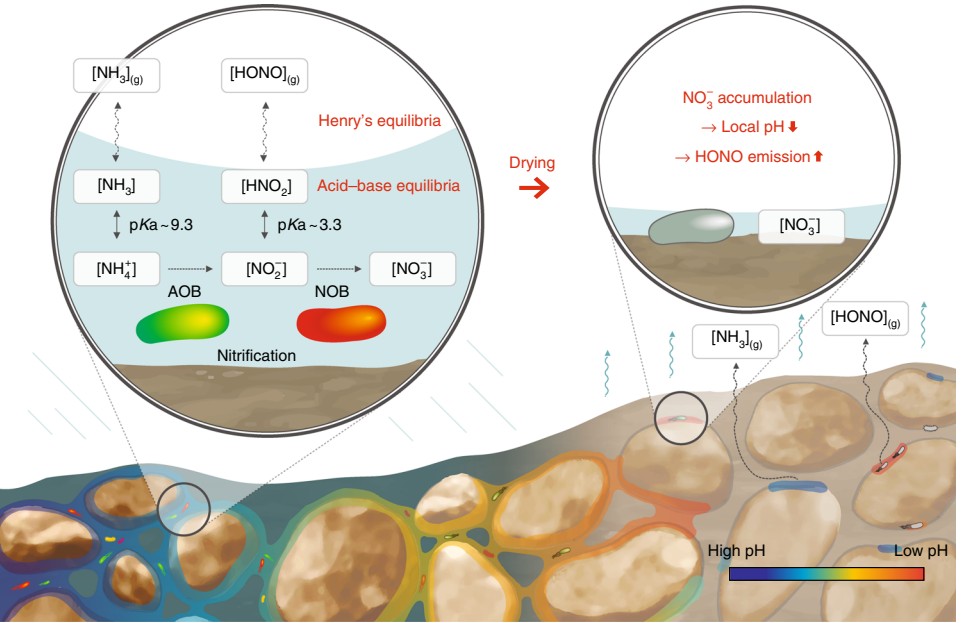

**Fig. 1** Nitrous acid and ammonia emissions from drying soils. A schematic of nitrous acid (HONO) and ammonia ($NH_3$) emissions due to biotic and abiotic processes in aqueous films on grain surfaces of unsaturated soil. Nitrification performed by ammonia oxidising bacteria (AOB) and nitrite oxidising bacteria (NOB) affects the gaseous emissions of $NH_3$ and HONO directly by altering the concentrations of their protonated forms within thin aqueous films. An increase in concentration during desiccation causes outgassing and precipitation of these compounds depending on their solubility. Their partitioning and chemical speciation are determined by Henry's law and acid–base equilibria. The product of aerobic nitrification, nitrate ($NO_3^-$), can accumulate locally and reach high concentrations that result in HONO emission hotspots with local acidity. This localised and highly dynamic process cannot be captured by averaged soil pH of saturated soils under static conditions

process. These observations raise several questions: First, the simultaneous HONO and $NH_3$ emissions from a soil or a biocrust appear to be in contradiction with the high levels of $NH_4^+$ and $NO_2^-$ and bulk soil pH in equilibrium. Second, the presence of nitrite oxidisers and the production of $NO_2^-$ by ammonia oxidisers (a direct source for HONO emissions) must be reconciled due to the expectation of $NO_2^-$ consumption by nitrite oxidisers. Finally, a prominent feature of HONO emissions from various soils while drying, points to a strong soil moisture dependency irrespective of nitrifiers' activity[23]. This dependency on soil hydration state motivated our interest in quantifying biotic and abiotic conditions in soil during drying. How could soil pH be affected by drying? How does microbial activity affect the aqueous phase chemistry of drying soils?

The effects of hydration dynamics on chemical and biological process at the microscale are poorly understood despite their ubiquity and potential importance for biogeochemical processes in surface soils. In this study, we employ a mechanistic model that integrates interactions between soil properties, microbial activity, and abiotic processes across air–water interfaces. We focus on the roles of hydration dynamics and the spatial heterogeneity of soil surfaces in modifying local pH related to gaseous emissions, especially HONO and $NH_3$ (Fig. 1). We first address general processes occurring within drying soils and demonstrate how the microscopic hydration conditions dictate the timescales of physicochemical processes that result in localisation of pH during drying. We then turn our focus to desert biocrusts that provide a case study of real soils and show how nitrifiers act as sinks and sources for modifying local conditions that can cause strong variation of pH within drying soils. The discussion follows with implications of our findings that highlight the general importance of hydration dynamics in determining gaseous emission of $N_r$, relevant for global N cycling.

## Results

**Water contents and water-film thickness distributions**. A quantitative description of soil gaseous exchange is strongly dependent on the representation of the soil aqueous phase and air–water interfaces. Macroscopically, soil hydration is often characterised by the water content and the matric potential, both are interdependent variables that modify gas diffusivity, aqueous phase connectivity and biological activity and thus gaseous fluxes from soil. The macroscopic representation, however, does not represent resolved geometrical information on soil aqueous phase at scales relevant to microbial life (submillimetre scales)[25,26] (a schematic of aqueous phase distribution in various hydration states is given in Fig. 2a). We thus employ a variable related to the water film thickness retained by rough soil surfaces to represent soil hydration status at the microscale and as the primary interface for gas uptake and emissions. The volume of the local water film also controls local concentrations of dissolved substances, thereby determining rates and amounts of matter exchange between gas and mineral surfaces.

We implemented a previously developed rough surface model[6,27,28] that links macroscopic soil water content to microscopic aqueous film thickness at different matric potential values (Fig. 2b). The film thickness reflects the amount of water retained to soil grain surfaces owing to the combined effects of adsorption and capillarity encapsulated in the definition of soil matric potential (energy state of soil water). The spatial heterogeneity of pore sizes and surface roughness yields a distribution of water film thickness across a soil domain that contributes to the macroscopic water content (for model details see Kim and Or[6,27]). Figure 2b shows that, as soil water content varies from about $0.3\,m^3\,m^{-3}$ (soil porosity) to about $0.01\,m^3\,m^{-3}$ (residual water content) during desiccation, the

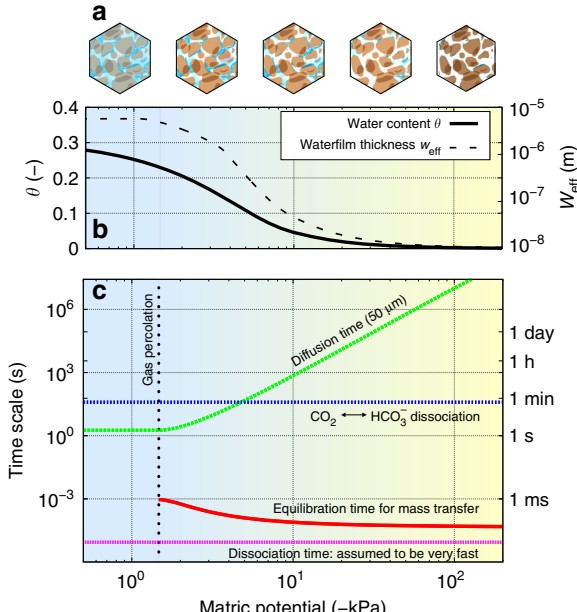

**Fig. 2** Changes in abiotic conditions while drying soils. Model predictions of changes in abiotic conditions during drying of soils. **a** A schematics of changes in aqueous phase configurations in soils during drying. **b** A typical model calculation of water content (black solid line) and effective water film thickness (black dashed line) as a function of matric potential (blue-yellow gradient represents relative wetness). **c** A comparison of characteristic timescales for physicochemical processes relevant for local pH determination in aqueous films for a range of hydration conditions (expressed as matric potential)

effective water film thickness (per unit soil surface area) varies by orders of magnitude from about $10^{-5}$ m at saturation to about $10^{-8}$ m (Fig. 2b in agreement with Tuller and Or[29]). This implies that the water loss at microscale cannot be scaled as the changes in water contents during desiccation.

**Timescales of physicochemical processes in unsaturated soils**. Changes in the distribution of aqueous film thickness during soil desiccation affect the timescales of various processes (Fig. 2c). Here, we focus primarily on physical and chemical processes within and across the gas–liquid interface. Near saturation (before gas percolation, marked as a vertical dotted line in Fig. 2c), water fills the soil pores and hinders gas exchange within the domain and nutrient diffusion and chemical processes are similar to aquatic systems. However, during soil desiccation, air percolates through empty soil pores and facilitates exchange of gaseous compounds to and from the residual water film on the rough soil grains. The large interfacial area of the thin water film in the soil matrix allows instant equilibration of mass transfer; thus, dissolved gases follow Henry's equilibria. Meanwhile, the aqueous diffusion becomes reduced under unsaturated conditions owing to lower connectivity and higher tortuosity of liquid phase[30]. Thus, lateral solute diffusion through the thin water film may become limiting relative to gaseous exchanges in unsaturated soils. The timescales of aqueous diffusion via thin films are estimated from $t \sim 2l/D_{eff}$ where $l$ is characteristic diffusion distance (50 μm as a representative local scale considering average intercell distances in soil is in the order of $10^{-5}$ m[31]) and $D_{eff}$ is the effective diffusivity of a solute at the given matric potential (Fig. 2c). Other chemical processes, such as acid–base dissociation or hydrolysis in water films are relatively fast and are assumed to instantly equilibrate in the model. This implies that the aqueous

diffusion becomes the most limiting step in terms of abiotic processes. Thus, this renders production and consumption of dissolved compounds that are highly localised under unsaturated conditions and gives rise to potential spatial heterogeneity in chemical conditions.

**Mean soil pH versus microscale aqueous film pH.** Changes in the aqueous phase configuration (i.e., film thickness distribution) in drying soils and gas phase percolation jointly shape concentrations of dissolved gaseous compounds, which are determined by mixing ratios of inorganic carbon and nitrogen (i.e., $CO_2$, $NH_3$, and HONO) based on Henry's law at local scale. Using these physical conditions, we calculated the local pH distribution of unsaturated soils by assuming acid–base equilibria and local charge balance (See Methods for details). Air–soil exchange and limited aqueous diffusion determine the spatial heterogeneity of pH within a drying soil even under the absence of biological activities. In addition, distribution of soil minerals such as iron, aluminium (hydr)oxides or calcite, also contribute to spatial heterogeneity of aqueous film pH at microscale[32]. We note that the reactivity of these minerals is also affected by hydration dynamics that determines dissolution of gaseous compounds (mainly $CO_2$). In the model, we consider a finite exchangeable $Ca^{2+}$ as a representative (calcite forming) mineral to mimic calcareous desert soils. $Ca^{2+}$ precipitation regulate the upper bound of local pH where a finite buffering capacity could be easily exceeded in shrinking aqueous volumes of locally isolated patch of water film during soil drying. A potential source of spatial variations in local pH is the distribution of chemical ions in aqueous phase, such as the highly soluble $NO_3^-$, that may be independent of gas phase constraints and could strongly affect local pH. We propose that nonuniform distributions of sources and sinks coupled with limited lateral diffusion in aqueous films may give rise to local imbalance in free cations and anions. Consequently, this affects local pH and results in strong spatial heterogeneity of pH (under unsaturated conditions) that would be difficult to reconcile with bulk soil pH measurements (for details, see Methods, Supplementary Figs. 1 and 2).

**Spatially resolved pH measurements of drying soils.** Evaporative water loss in soils increases concentrations of chemical compounds and precipitation of salts. These changes influence acid–base dissociations that are kinetically rapid and highly depend on pH of aqueous solutions. For systems with limited buffering capacity, pH is likely to vary during soil desiccation. Surprisingly, such a local and dynamic aspect has been missing in studies that often consider a constant bulk soil pH value irrespective of hydration conditions.

To examine the dynamic and local nature of soil pH during drying, we conducted a series of proof of concept tests by measuring the pH of buffer solutions and wet quartz sand (sterilised) under two wet–dry cycles (see Methods, Supplementary Figs. 3 and 4). The primary objective of this simple test was to illustrate the abiotic mechanism for the onset of pH zonation during soil drying at the microscale. We opted for using a simple system to avoid complexities of natural soils with poorly defined composition and unconstrained microbial activity that would require dedicated experimental setups to evaluate the far more complex role of the microbial component of the phenomenon. The pH values and maps were obtained from planar pH optodes (PreSens GmbH, Rosensburg, Germany)[33] and simultaneously verified using independent measurements with microelectrodes (PH-200C, Unisense, Aarhus, Denmark) (Fig. 3a, Supplementary Fig. 4b, c). Optode measurements exhibited a consistent (albeit mild) decrease in pH (Fig. 3b, d magenta and purple lines about

0.2–0.3 units with optode precision of 0.01 pH unit at pH 7) during drying that lends support to the hypothesis that evaporation alters the pH in the remaining water films (changes in hydration conditions are given in Fig. 3c, e). The pH electrodes revealed a more drastic drop of about 1 pH unit (Fig. 3b, d, turquoise and orange lines). This suggests that the buffering capacities of pore water and the solution were exceeded in the remaining small volumes of aqueous films. The differences in the magnitude of pH values measured by the optodes and the electrodes may also reflect on the nature of the measurement itself (highly localised with the electrodes and more diffusive with the optodes).

The optodes not only allowed for observations to dry conditions (dryer than possible with the electrodes), but they also provided a spatial distribution of pH values. We have used sand of different textures (different surface areas and retained water films) and modified $pCO_2$ levels in the injected air into the measurement cuvette (Fig. 4). In these measurements, the sample of sterilised quartz sand formed two distinctive regions with fine and coarse grain sizes to accentuate the nonuniform pH dynamics during drying. The nested behaviour in spatially averaged pH of the entire region highlighted relations between local pH and local soil textures. This relation persisted under different $pCO_2$ levels in the air suggesting a potential role of soil microscale structure affecting local pH dynamics (as also demonstrated by the vertical gradient of pH during drying; Supplementary Fig. 5). Furthermore, increasing $pCO_2$ levels increased the concentrations of carbonic acid and lowered the pH of the entire domain (Fig. 4c).

These results, should be interpreted with caution because the responses of the optode and electrodes were designed primarily for wet conditions, hence we trust results from intermediate hydration conditions where the optode remains fully hydrated (while film diffusion becomes limited). These limitations notwithstanding, these preliminary measurements demonstrate how local pH varies during soil desiccation.

**Predicting HONO and $NH_3$ emission from drying biocrusts.** The discussion is extended from drying sterile soil to a soil system with bacterial communities. There is no doubt in microbial activities in soils would intensify the spatial heterogeneity of localised film pH in drying soils. This is because they are the active sources and sinks of various substances including gaseous compounds like $CO_2$. In this work, we show that the microbial activity within soil causes the pH zonation during drying that would trigger the concurrent emission of HONO and $NH_3$. For this, we have employed a previously developed mechanistic model of biocrust[6], which describes the activity of an established microbial communities, including nitrifiers, ammonia oxidisers (here, noted AOB), and nitrite oxidisers (NOB) together with other members such as phototrophs, heterotrophs, and denitrifiers. For comparison of model predictions with laboratory measurements using real biocrusts[19,34], we considered a wetting-drying event applied to model biocrust under darkness (hence no photosynthesis) mimicking conditions of reported lab experiments[18,19,34].

Figure 5 summarises the simulated dynamics of drying biocrusts. During the 24 h of simulated drying (Fig. 5a), the net biogenic production rate of soil $NO_2^-$ was negative due to the consumption rates by NOB exceeding production rates by AOB (Fig. 5b). Consequently, microbial activity (combining AOB and NOB) did not provide a direct source for HONO emissions in this case (the system acted as a sink of HONO via Henry's law). The strong variations in local pH resulted from the joint effects of microbial activity and desiccation (Fig. 5c). Under wet conditions (high saturation), most of the domain is alkaline (and the bulk pH

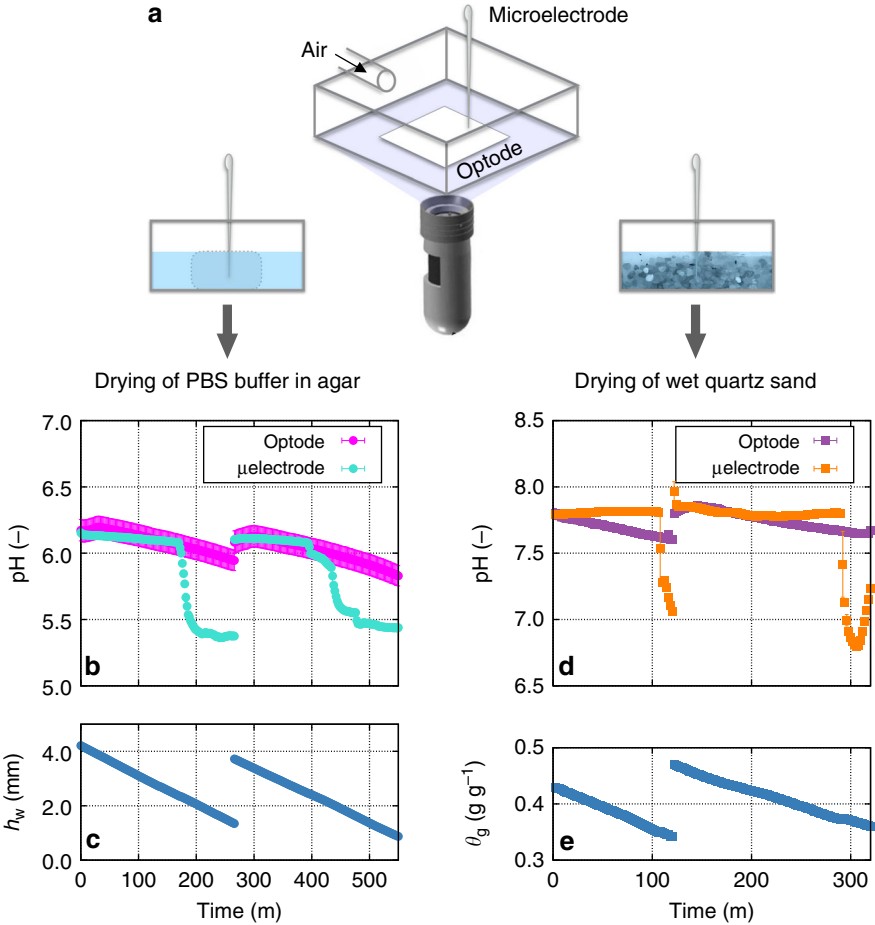

**Fig. 3** Measurements of local pH under wet–dry cycles. Laboratory measurements of pH dynamics under two wet–dry cycles monitored using a planar pH optode (pH sensor SF-HP5-OIW, PreSens GmbH, Rosensburg, Germany) and a pH microelectrode (PH-200C, Unisense, Aarhus, Denmark). **a** An illustration of the measurement cuvette and experimental setup. The optode imaging sensor was mounted at the bottom of the glass cuvette and the microelectrode was installed from the top, upright. A small glass cuvette (20 mm × 20 mm × 20 mm) filled with an agar block saturated with phosphate buffer saline (PBS pH = 6.1, 0.1 M) (left) or wet quartz sand (right) while monitoring pH variations during drying. Sample desiccation was controlled by airflow rate (relative humidity 20%) into the cuvette and hydration status of the sample was monitored simultaneously by weighing the entire sample. **b** pH changes in drying agar monitored with the optode (magenta circles) and the microelectrode (turquoise circles). Here, optode measurements were spatially averaged (pixels within an area of 25 mm$^2$) and electrode measurements were temporally averaged (measured at 5 s interval and avearged for 1 min). Error bars indicate ±1 standard deviation. **c** The amount of water in the cuvette was measured in weight and the value was translated to equivalent water depth of the agar cube (maximal value was 4 mm). **d** pH changes during drying of wet quartz sand monitored with the optode (purple squares) and the microelectrode (orange squares). **e** variations in the hydration status of the sand expressed as gravimetric water contents (weight of water/weight of wet sand [g g$^{-1}$])

is near 7), thus high levels of NH$_3$ volatilisation occurred at the soil surface (marked by a positive-NH$_3$ flux in Fig. 5d). The emission of NH$_3$ increased following desaturation and invasion of gas phase through the domain (marked by gas percolation degree in Fig. 5a, and dotted arrow on the left side). These reflect an impediment to gas emissions under high saturation irrespective of local chemical conditions. Furthermore, simulations show a decrease in aqueous film pH during drying similar to observations (Figs. 3 and 4). The resulting spatial variations in local pH span a range of pKa values for HONO with an increase in emission rates (Fig. 5c–f). The local acidification of the water film drives HONO release and NH$_3$ absorption. Following complete desiccation of the biocrust with the cessation of biological activity and high local acidification, HONO efflux proceeds abiotically as outgassing by Henry's law and volatilisation (Fig. 5d green line).

We attribute this local acidification to nitrification that results in accumulation of NO$_3^-$ while water is removed by evaporation

(Fig. 5e, f). To examine effects of hydration conditions and local nitrate accumulation on aqueous film pH, we systematically calculated local pH as a function of nitrate amounts and matric potentials (Fig. 5g). For this calculation, we ignored diffusion across aqueous patches and consider evaporative concentrations and instantaneous equilibrium of gas–liquid partitioning at local scale only (the size of a connected liquid patch is of the order of 100 μm$^2$). Results suggest that the local amount of NO$_3^-$ is the primary determinant of local pH during evaporative water loss because other inorganic components (carbon and nitrogen) are constrained by the protonated forms of their gaseous compounds (e.g., NH$_3$ + H$^+$ ⇌ NH$_4^+$, HONO ⇌ NO$_2^-$ + H$^+$).

**Characteristics of HONO and NH$_3$ emissions.** Measuring microscale soil pH heterogeneity and separating abiotic and biotic effects under unsaturated conditions remain a challenge. We thus use the model to systematically evaluate HONO emissions under

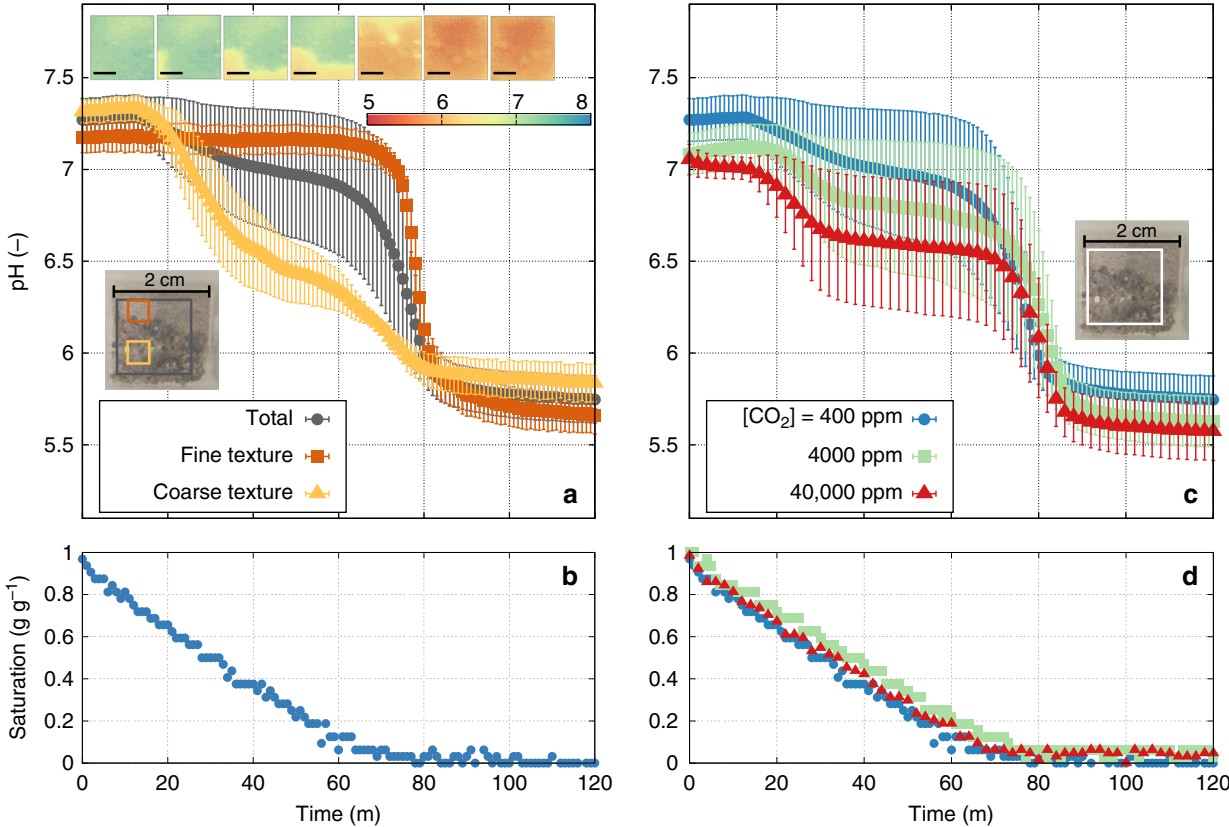

**Fig. 4** pH localisation and dynamics during drying of quartz sand. Direct measurement of pH localisation and dynamics during desiccation of gamma-ray sterilised quartz sand of different textures under different partial pressure of carbon dioxid ($pCO_2$). **a** pH dynamics was analysed by averaging differnt regions with fine texture (0.08–0.2 mm, the red box in the inset image of a top view of the sample), coarse texures (0.7–3 mm, the yellow box in the inset), and the entire domain (labled as total, the grey box in the inset). Spaitally averaged pH values for each region are shown as symbols with error bars (±1 standard deviation). On the top of this figure, the dynamics of spatial pH maps during pH transition are given at 20 min intervals (the scale bar indicates 5 mm). **b** The saturation dynamics during evaporation defined as the amount of water in the sample relative to the amount of deionised water applied for saturating the sample). **c** The variations in spatially averaged pH of the same sterilised quartz during drying for different levels of $pCO_2$ in the measurement cuvette. Error bars indicate ±1 standard deviation of the entire region (the white box in the inset image of a top view of the sample). **d** Saturation dynamics during desiccation for experiments conducted under different $pCO_2$ levels

a range of conditions including different drying rates and atmospheric $NH_3$ levels.

Desiccation rates regulate the optimal time window for HONO and $NH_3$ emissions (Fig. 6a–c, dotted lines for slow drying and solid lines for fast drying, Supplementary Fig. 6a–c) through their joint dependency on water contents and pH (Fig. 7 and Supplementary Fig. 7). Simulations suggest the $NH_3$ emissions to occur before HONO emissions during a course of drying. In addition, the absorption of $NH_3$ to water film is expected at the peak of HONO emission illustrating the interrelation between these two gases that are mediated by local pH in the aqueous phase. The mixing ratios of these gases in the air affect magnitudes of HONO emission and $NH_3$ absorption during drying. Higher $NH_3$ levels increases the maximum emission flux of HONO by promoting AOB activity with higher nitrification rates (See Fig. 6 and Supplementary Fig. 6d–f).

The water content dependency of gaseous emission is illustrated by plotting the simulated emissions as a function of percentage of water-holding capacity together with spatial variance of local pH values (Fig. 7a, b, Supplementary Fig. 7). In Fig. 7, we compare model simulations with HONO emission rates reported in laboratory studies of cyanobacterial biocrusts (without higher organisms such as moss or lichens)[19,34]. The comparison shows that the model captures the salient features of biocrust HONO emissions, with the characteristic single peak at

"optimal" water content (for different drying rates and atmospheric $NH_3$ levels). We note that the peak HONO emissions does not occur at the same water content for all conditions (although the range is narrow 10–25% of WHC). The results suggest that the desiccation rate affects the shape of the HONO emission peak, and these drying patterns reflect the properties of the biocrust and external driving forces (evaporation rate). In addition, the level of atmospheric $NH_3$ determines the magnitude of the HONO emission peak and these can be related to the input of nitrogen to the domain, such as activity of diazotrophs in the microbial community or external input by atmospheric deposition or application of fixed nitrogen.

## Discussion
The results of our simple experiments using sterilised sand under laboratory conditions (Figs. 3 and 4) and the mechanistic biocrust model simulations (Figs. 5–7) confirm the onset of spatially localised processes emerging in drying soil and biocrusts. Local variations in the concentrations of dissolved substances during desiccation induce changes in local pH at microscale that, in turn, give rise to hotspots for emissions of pH-dependent gases such as HONO and $NH_3$ (Fig. 7). Although our laboratory tests were performed under simple conditions of sterile soil, they provide a direct proof of concept for these processes. The prediction by the mechanistic model, the onset of microscale processes under

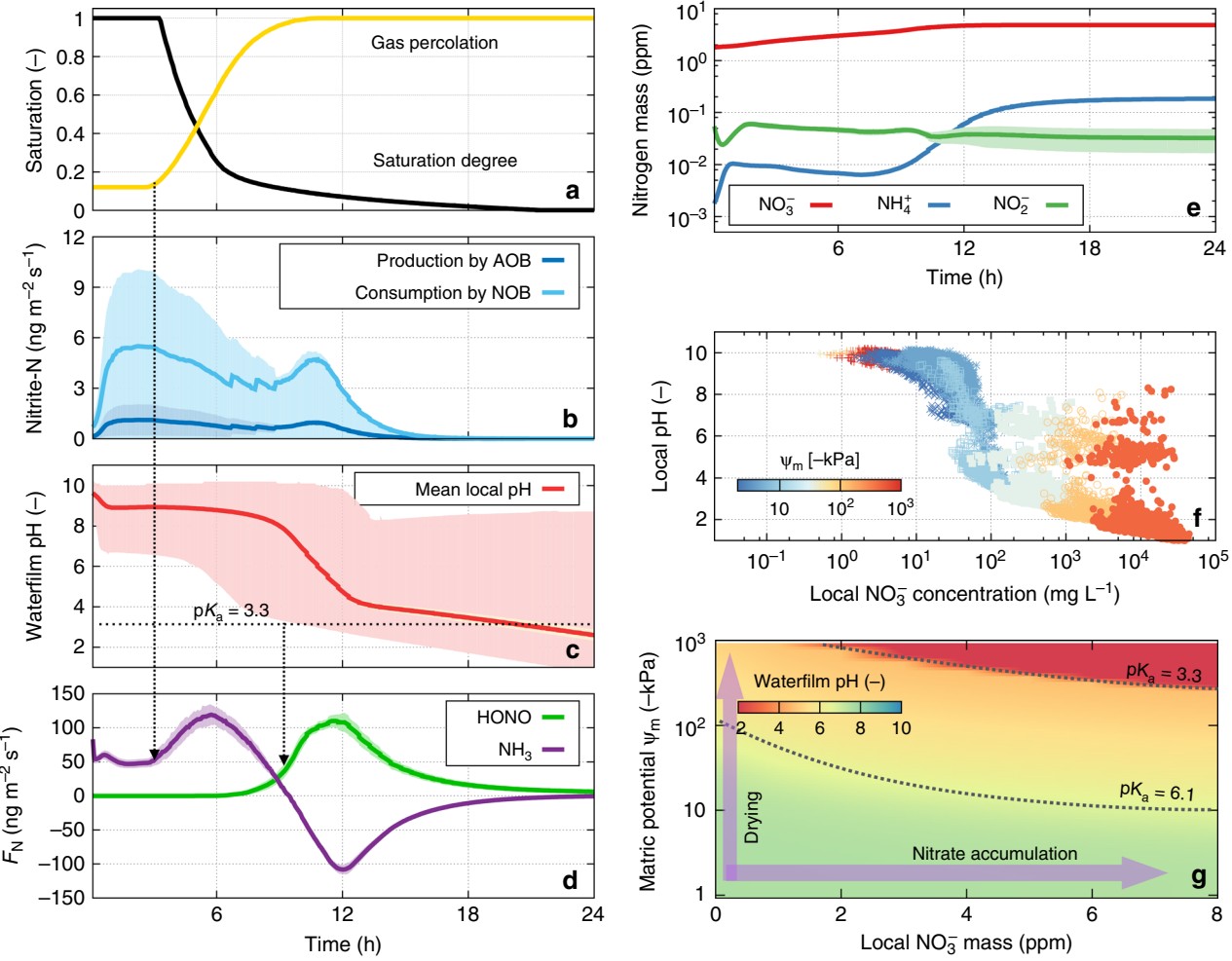

**Fig. 5** Predicted dynamic processes during biocrust desiccation. Dynamic processes during biocrust desiccation as predicted by the desert biocrusts model (DBM). The results were obtained from eight different simulations with identical boundary conditions. **a** Changes in saturation and increase in gas percolation during 24 h drying. **b** Simulated production and consumption of nirite ($NO_2^-$) by microorganisms; $NO_2^-$ consumption by nitrite oxidising bacteria (NOB) (light blue) exceeds the production by ammonia oxidising bacteria (AOB) (dark blue). Solid lines are the averaged values and shaded areas indicate 1 standard deviation of all simulations ($n = 8$). **c** Mean local pH (red line) where spatial heterogeneity of local pH spans a wide range of pH values (shaded area indicates from the minimum to the maximum). **d** The dynamics of nitrous acid (HONO) (green) and ammonia ($NH_3$) (purple) emissions from the model biocrust with positive- and negative-flux values indicate emission or uptake by the domain, respectively. **e** Simulated variations in inorganic nitrogen compounds with nitrate ($NO_3^-$) (red), ammonium ($NH_4^+$) (blue), and $NO_2^-$, (green) during drying (values are given in ppm with the unit $g\,g^{-1}_{soil}$). **f** Simulated local concentrations of $NO_3^-$ in the aqueous phase plotted against local pH at 4 h intervals ($t = 0, 4, \cdots, 24$) during drying (the colour bar corresponds to the matric potential, $\psi_m$, and the values are of the simulations). **g** The relationship between local aqueous film pH as a function of hydration state (expressed as matric potential) and the amount of $NO_3^-$ in ppm

dynamic hydration conditions, was partially confirmed using a simple experimental system and two independent measurement methods. We have not pursued more complex experimental systems (biocrust or agricultural soils) due to the steep increase in characterising and taming natural complexity at this preliminary phase. We thus focused in this study on the proposed mechanism of pH localisation and its potential consequences using a mechanistic model and provided preliminary laboratory measurements that support the proposed mechanism. The study demonstrates that macroscopic metrics such as mean water content and bulk soil pH, may not capture the nuances associated with efflux patterns such as HONO emissions from alkaline soils or the concurrent emissions of $NH_3$ and HONO during cycles of wetting and drying. The results suggest that spatial variations at microscale are critical and inclusion of hydration dependency of aqueous film thickness and local pH distributions as essential ingredients for the observed emissions.

The simulated microbial activities in the model show that $NO_3^-$ accumulated in thinning water films that act as a driver for pH zonation. Locally accumulated $NO_3^-$ seems to control the changes in local pH drastically (Fig. 5g) because of its high solubility (in the range of $\sim 10-1000\,g\,L^{-1}$), and because it can be protonated to nitric acid ($HNO_3$) only under extremely acidic conditions ($pKa \sim -1.4$). This implies that the localised sources or sinks of $NO_3^-$ during evaporative water loss under diffusion limitation give rise to strong heterogeneity in pH covering both pKa values for HONO and $NH_3$.

The most dominant sink and source of $NO_3^-$ in soil is nitrification and denitrification resulting from microbial activity. Considering that nitrification is a strictly aerobic process and desiccation will oxygenate most of near-surface soils, accumulation of $NO_3^-$ is likely to happen (see the simulated distributions of $O_2$ and $NO_3^-$ in Fig. 5 of Kim and Or[6]). This accumulation could be responsible for the large HONO emissions observed in desert

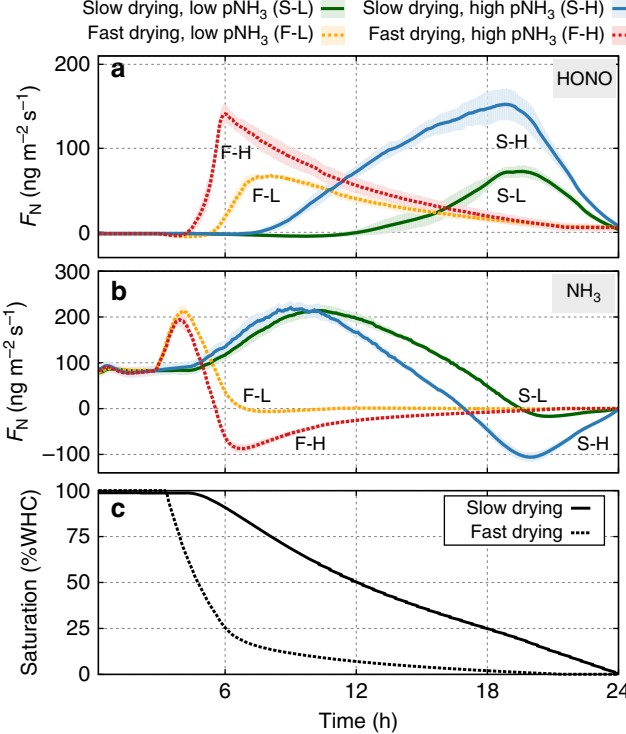

**Fig. 6** Nitrogen gaseous emission dynamics under various conditions. Nitrous acid (HONO) and ammonia ($NH_3$) gaseous emissions during biocrust drying as a function of time. Typical simulations of different conditions in drying patterns (solid lines: slow drying, dashed lines: fast drying), and atmospheric $NH_3$ levels (low: 5 ppb, high: 20 ppb) are denoted as slow/high drying with low-/high-$NH_3$ level, S–L (green), S–H (blue), F–L (orange), and F–H (red), respectively. The simulated dynamics of **a** HONO and **b** $NH_3$ emissions are plotted during **c** 24 h of drying at two rates. Hydration conditions are expressed in percent of water-holding capacity. Lines are averaged values of simulations ($n = 8$) and shaded areas indicate ±1 standard deviation

biocrusts. The observed strong correlation between the emitted amounts of HONO and high nitrification rates[24] or high contents of $NO_3^-$ and $NO_2^-$[19] support our hypothesis of local acidification due to $NO_3^-$ accumulation during soil drying.

However, the measurements in lichen- and bryophyte-dominated biocrusts[18] did not show a strong correlation with $NO_3^-$ accumulation. These biocrusts emit smaller amounts of HONO over a wider range of water contents unlike the well-defined peak of HONO emission from cyanobacterial crusts[18,34]. We attribute this to the characteristics of well-developed biocrusts with larger organisms that could be the major sources/sinks of $N_r$ compounds. In addition, well-developed biocrusts have a higher content of extracellular polymeric substances (EPS) that would further distinguish such biocrusts from other soils or other bio-crusts in their initial phase. Higher-EPS contents (such as in active biocrusts) are likely to modify local hydrology owing to the increased water-holding capacity, lower-hydraulic conductivity, and potential delay of evaporative drying rates[35–37]. We note that the presented model includes the production and accumulation of EPS, but swelling–shrinking dynamics or its dehydration processes are not considered due to their complex dependency on the amount of EPS, pH of the soil solution, temperature, presence of cations, etc[6]. To keep the model in this study as simple as possible, we include the potential impact of EPS as a modification in drying rates (Figs. 6 and 7, the presence of EPS analogous to slow drying, S). Results suggest that slower rates of desiccation may

lead to broader peaks of HONO emissions due to the increased activity of nitrifiers. However, we note that the relations can be more complicated due to effects of delayed desiccation on deni-trifiers or nitrate reducers and their ability to consume $NO_3^-$ under anoxic conditions[38,39]. This indicates the importance of oxygen distribution within drying soils and the interplay between aerobic nitrifiers and anaerobic denitrifiers as sinks/sources of $NO_2^-$ and $NO_3^-$[38,40,41]. Here, we focused primarily on aerobic processes, such as nitrification, since desert and near-surface soils are often dry and mostly aerated (shown in Fig. 5 of Kim and Or[6]). Furthermore, we also observed that, in our simulations, the heterotrophic activity of denitrifiers was inhibited due to the limited extent of anoxic regions and the absence of carbon sources, notwithstanding their presence in the model. In other soil systems with sufficient carbon sources, the presence of anoxic or anaerobic microsites is an important factor even near the soil surface especially when oxygen consumption by aerobic organ-isms and shallow roots may exceed its diffusion rates into the soil[42]. The conditions within soil aggregates and in fine textured soils with appreciable EPS promote the formation and persistence of anoxic microsites, that, in turn, may affect N-losses following wetting due to anaerobic production of $N_2O$ for instance[5,43,44]. We should mention that the model does not include other pathways, such as nitrifier denitrification[45] or nitrate ammonifi-cation[46] that produce $N_2O$ or NO and could affect the estimation of N gaseous effluxes reported in this study.

The model results suggest that higher atmospheric $NH_3$ levels could increase $N_r$ losses via HONO emission (Figs. 6 and 7, denoted as H). The positive net $N_r$ emission during wet–dry cycles indicates that $NH_3$ absorption at low-water contents can-not compensate the gaseous loss. The atmospheric $NH_3$ input to the soil can be interpreted as an additional $N_r$ source resembling agricultural input or cultivation induced biological nitrogen fixation. Thus, we argue that higher input of $N_r$ to nitrifying communities in soils would trigger increased $N_r$ loss to the atmosphere after every cycle of drying and wetting. This implies a strong dependency of $N_r$ gas emission (HONO and $NH_3$) and solute accumulation ($NO_3^-$) on precipitation frequency and soil structure regardless of additional factors (e.g., EPS content or the distribution of oxic and anoxic zones).

So far, we have shown the pH zonation in shrinking water films during drying acts as a trigger for HONO and $NH_3$ emissions. The model of desert biocrusts enabled us to explore underlying mechanisms due to the explicit representation of the microbial community and the distribution of their functional members[6]. Although tested on biocrusts, we argue that the pH zonation mechanism for HONO emission is generally applicable to any soils since it is caused by orchestrated activities of ubiquitous nitrifiers and abiotic processes under evaporative forcing. We presented measurements of local pH on sterilised sand as a proof of concept that lays the ground for further experimentation using real soils with intact microbial communities. The mechanistic model was instrumental in elucidating the puzzle of concurrent gaseous emissions (HONO and $NH_3$), yet various aspects of the model can be developed further such as realistic representation of all aspects of EPS (hydration to diffusion effects). An interesting and high priority addition would be the inclusion of a recently discovered pathway using NO as an obligatory nitrification intermediate[47], considering such pathway could shed light on similarity of emission patterns of NO and HONO from drying soils. Furthermore, it would help quantifying abiotic NO emis-sions during drying that could affect the activity of NOB thus modify nitrification rates and accumulation of $NO_3^-$. A natural extension of this study is to consider agricultural soils and sup-port recent findings of anaerobic nitrate reduction in oxygen-limited microsites that act as a source of HONO under wet

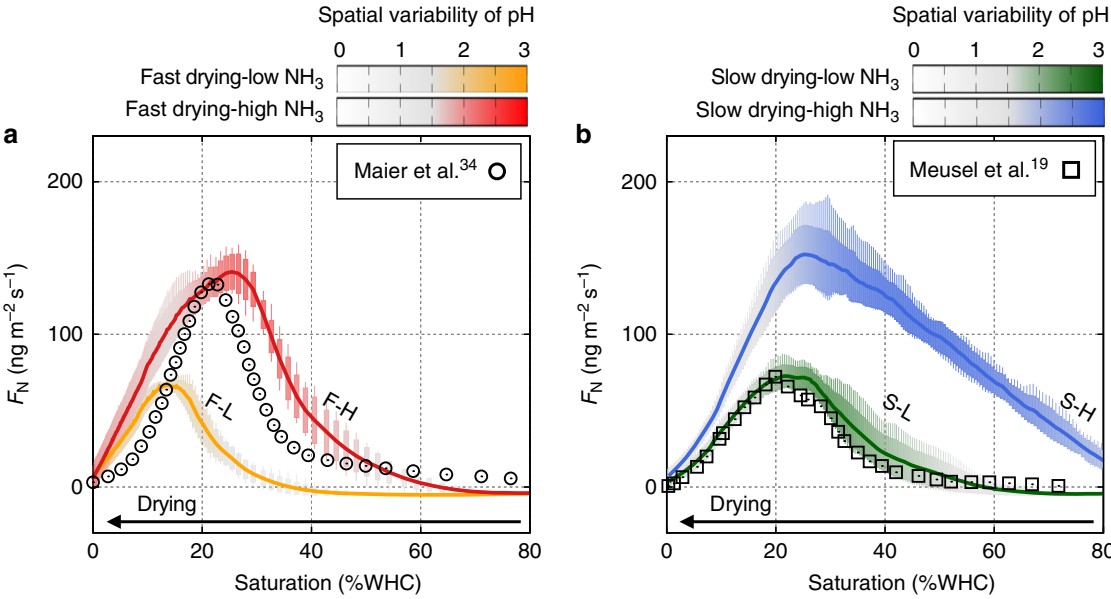

**Fig. 7** Model prediction and observation of nitrous acid emission. Nitrous acid (HONO) gaseous emissions during biocrust drying as a function of soil hydration conditions (expressed in percent of water-holding capacity). Typical simulations of different conditions in drying patterns, and atmospheric ammonia (NH₃) levels (low: 5 ppb, high: 20 ppb) are denoted as slow/high drying with low-/high-NH₃ level, S–L (green), S–H (blue), F–L (orange), and F–H (red), respectively. The length of each box indicates ±1 standard deviation and each stick ranges the minimum and maximum emission of HONO from all simulations (n = 8). Colour gradients indicate the averaged spatial variability of local pH values (given as ±1 standard deviation) across simulations. **a** Simulated HONO emission with fast drying under high NH₃ input was comparable with measurements from cyanobacteria-dominated crust in South Africa[34]. **b** Simulated HONO emission with slow drying under low NH₃ input was comparable with measurements from light crust in Cyprus[19]

conditions[39]. Such model refinements would enhance our understanding of general mechanisms dictated by microscale processes with respect to the factors shaping them, as shown for pH zonation driven by dynamics of soil hydration. Ultimately, this could lead to improved predictions of nitrogen partitioning between soils and the atmosphere; a highly relevant aspect for regional and global models of the nitrogen cycle.

## Methods

**Mathematical model of desert biocrusts.** The desert biocrust model (DBM)[6] is a mechanistic model that links the aqueous state with geochemical processes and biological activity in pioneer desert biocrusts (no lichens and mosses). The DBM considers diffusion-reaction, mass transfer at gas/liquid interface, and chemical processes like C and N dissociation, volatilisation, and precipitation, whereas microbial processes are described by an individual based representation of cells. The biocrust microbial community consists of four functional groups; photo-autotrophs, aerobic heterotrophs, denitrifiers (anaerobic heterotrophs), and chemoautotrophs (nitrifiers; AOB and NOB). The cycles of carbon and nitrogen are performed only by microorganisms (no higher organisms) and thus representing cyanobacteria-dominated biocrusts. For fully saturated biocrusts, the model has been tested extensively and found to agree with multiple lab experiments in terms of dynamics of oxygen and pH profile, and CO₂ efflux from biocrust under day–night cycles[6]. This study extends the previous work by exposing the microbial community to dynamic hydration conditions (wet–dry). In other words, we have used the distribution and abundances of microorganisms obtained at full saturation as initial conditions for the subsequent desiccation and rewetting cycles. We note that the simulations mimicked the "darkness" of the lab conditions, where HONO emission dynamics were measured[18,19,34]. Therefore, there was no photosynthesis during drying and the activities of chemoautotrophs as nitrifiers governed the gas emission dynamics. In this study, the atmospheric level of HONO was kept constant as 1 ppb in agreement with field measurements for semiarid pine forest[21]. The mixing ratio of NH₃ was used as a control parameter for the simulations of Figs. 6 and 7. In Supplementary Fig. 6, we varied the atmospheric level of NH₃ from 0.1 to 10 ppb (representing typical values that are in the range of 1–10 ppb depending on the time of the day, season, and regions). Detailed description is provided in Kim and Or[6]. In the following sections, we summarise some of the important model description that are relevant for this work.

**Diffusion processes in the DBM.** A spatially explicit model of soil profile was employed to estimate the gaseous efflux. The domain is discretised as a set of spatial elements (hexagonal patches), which represents a local property of soil

structure, such as porosity and surface roughness following the approach of the modified rough surface patch model[6,27,28]. At a given hydration condition, local water/gas contents can be determined and these measures shape apparent diffusion rates and mass transfer rates at the gas–liquid interface. As a collection of patches, percolation of gas phase was considered to achieve substrate concentration profiles at top soils. Supplementary Fig. 1a shows an example of physical domain of the model, including a schematics of the soil vertical domain as a cross-section, gas contents distribution, and the percolating pathway of the gas phase. Gas diffusion coefficient is about 10⁴ times higher than the diffusion process at the bulk liquid. Thus, we assume that at top soils (near the surface, within a few cm scale), patches that are connected to the air keep the constant mixing ratios (partial pressures); therefore, the air in our model is assumed to be an infinite source. In this work, we only considered the temperature-dependent air density which determines the concentration level of each gaseous substrates. Various reactions, such as photo-enhanced reactions and diurnal cycles of its rate, are not considered in this model. We focused on the production of gaseous elements from soils and its release to the air from the surface. The diffusion process of a substrate with the concentration $C$ at the aqueous phase is described as following

$$\frac{\partial C(\vec{r}, t)}{\partial t} = \nabla \cdot (D(\vec{r}) \nabla C(\vec{r}, t)) - \text{sink terms} + \text{source terms} \quad (1)$$

where $D(\vec{r})$ is the apparent diffusion coefficient defined from the effective film thickness distribution of adjacent patches and the effective diffusion coefficient including tortuosity as a function of porosity and water content using Millington–Quirk equation[48].

$$D_{\text{eff}}(\vec{r}) = D_0 \frac{\theta(\vec{r})^2}{\phi(\vec{r})^{4/3}}, \quad (2)$$

where $D_0$ is the diffusion coefficient at the bulk liquid, $\theta(\vec{r})$ and $\phi(\vec{r})$ represent the local water content and the local porosity, respectively. Thus, the net flux between two adjacent patches (for example, patch 1 and patch 2) due to diffusion is calculated as following

$$\vec{J}_{1 \to 2} = -\frac{2D_{\text{eff}}(\vec{r}_1)D_{\text{eff}}(\vec{r}_2)}{D_{\text{eff}}(\vec{r}_1) + D_{\text{eff}}(\vec{r}_2)} \min[d_w(\vec{r}_1), d_w(\vec{r}_2)] \frac{C(\vec{r}_2) - C(\vec{r}_1)}{|\vec{r}_2 - \vec{r}_1|}, \quad (3)$$

where $d_w$ indicates the effective water film thickness. To calculate the flux between heterogeneous medium, we have chosen the harmonic mean of diffusion coefficient and the minimum value of water film thicknesses between neighbouring patches (for the details, see Kim and Or (2016)[27]). In Eq. (1), the second and third terms on r.h.s. indicate sinks and sources of the substrate as reaction terms. Source and sink terms are the mass transfer from air to soil water, the net consumption by microorganisms, and the compensation for ionic charge neutrality from chemical reactions. Essentially, all biological, chemical reactions occur within the aqueous

phase and distribution of substrates are assigned only with diffusion and mass transfer between gas and liquid phases.

**Gas–liquid partitioning under Henry's law.** In this work, following gas and liquid partitionings are included

$$O_2(aq) \rightleftharpoons O_2(g) \tag{4}$$

$$CO_2(aq) \rightleftharpoons CO_2(g) \tag{5}$$

$$HNO_2(aq) \rightleftharpoons HONO(g) \tag{6}$$

$$N_2O(aq) \rightleftharpoons N_2O(g) \tag{7}$$

$$NH_3(aq) \rightleftharpoons NH_3(g). \tag{8}$$

Unsaturated soils are characterised with the large specific surface area with the thin water film thickness, thus the mass transfer can be assumed to be a very fast process and the concentration of two phases can be obtained with Henry's equilibria

$$C^l = H_{cc}(T)C^g = k_H(T)P_g, \tag{9}$$

where $C^l$ and $C^g$ are concentrations in liquid and gas phases, respectively. $H_{cc}(T)$ is the dimensionless Henry's constant at temperature $T$; $H_{cc} = H_{cc}^\Theta e^{-\frac{\Delta_{soln}H}{R}\left(\frac{1}{T} - \frac{1}{T^\Theta}\right)}$, where $\Delta_{soln}H$ is the enthalpy of solution, $R$ is the gas constant, and the superscript $\Theta$ refers to the standard condition ($T^\Theta = 298.15$ K)[49]. The Henry's constant, $k_H$, is defined with the partial pressure of the gaseous compound, $P_g$. The partial pressure (mixing ratio) of each chemical compound in gas phase is important as it acts as a constant boundary condition in the model. Unlike the atmospheric chemistry, that aims at calculating the equilibrium gas phase concentration from the soil pH value[21], this proposed work focuses on the soil pore water and its local chemical reactions by assuming that the air as a well-mixed infinite source. We note that determining atmospheric level of trace gases can be tricky as it varies depending on locations and conditions when it measured. For example, the atmospheric HONO concentrations from the field measurement range from tens of parts per trillion to several parts per billion[21]. In this work, some values from literatures are selected as the fixed atmospheric equilibrium and used to calculated the gaseous efflux. Used values can be found in Supplementary Table S1 of Kim and Or[6].

**Acid–base model and local pH determination.** Coupled ODEs of chemical acid–base reactions are used to determine chemical status of local water film. Local pH was calculated after net diffusions and microbial reactions with an assumption of local charge neutrality. Considered reactions in aqueous phase are

$$H_2O \rightleftharpoons OH^- + H^+ \tag{10}$$

$$CO_2(aq) + H_2O \rightleftharpoons HCO_3^- + H^+ \tag{11}$$

$$CO_2(aq) + OH^- \rightleftharpoons HCO_3^- \tag{12}$$

$$HCO_3^- \rightleftharpoons CO_3^{2-} + H^+ \tag{13}$$

$$NH_3(aq) + H^+ \rightleftharpoons NH_4^+ \tag{14}$$

$$HNO_2(aq) \rightleftharpoons NO_2^- + H^+ \tag{15}$$

$$CaCO_3(aq) \rightleftharpoons Ca^{2+} + CO_3^{2-}. \tag{16}$$

The detailed rates and the acid–base dissociation constants are given in Supplementary Table S2 of Kim and Or[6]. Concentration of protons, pH, was locally determined with an assumption of local charge balance via self-ionisation of water

$$[H^+] + [NH_4^+] + 2[Ca^{2+}] + [Z^+] - [OH^-] - [HCO_3^-] - [NO_2^-] - [NO_3^-] - 2[CO_3^{2-}] = 0, \tag{17}$$

where $[H^+][OH^-] = K_W = 10^{-14}$. By solving the differential algebraic equations, local pH values were obtained during the entire dynamics. The unknown cation, $Z^+$, is considered to be nonreactive, but it is added to the system for charge compensation for pH value of real soil with various minerals. In addition, calcium ($Ca^{2+}$) is selected as a representative reactive cation that participates for inorganic carbon availability. All chemical reactions are coupled with the availability of protons at a given temperature. During the acid–base calculation, the extended Debye–Hückel equation is used for all ionic interactions. Supplementary Fig. 1a shows a schematic of the local pH calculation and a typical result of the calculation in Supplementary Fig. 1b.

**Analytic solution of local pH under Henry's and acid–base equilibria.** At the interface between the air and soil water, the fast equilibration of chemical processes can determine local pH of water film. At equilibrium, constraints of gaseous

compounds via Henry's law and the acid–base dissociation relations lead to

$$k_H^C P_{CO_2} = [CO_2] = \frac{[HCO_3^-][H^+]}{K_{1c}} \tag{18}$$

$$[HCO_3^-] = \frac{[CO_3^{2-}][H^+]}{K_{2c}} \tag{19}$$

$$[CaCO_3] = \frac{[Ca^{2+}][CO_3^{2-}]}{K_{sp}} \tag{20}$$

$$[NH_4^+] = \frac{[NH_3][H^+]}{K_A} = \frac{k_H^A P_{NH_3}[H^+]}{K_A} \tag{21}$$

$$k_H^N P_{HONO} = [HONO] = \frac{[NO_2^-][H^+]}{K_N}, \tag{22}$$

where $k_H^C$, $k_H^A$, and $k_H^N$ are Henry's constants of $CO_2$, $NH_3$, and HONO, respectively. Dissociation constants are denoted with capital $K$ for each acid–base reaction. We note that Henry's constants and dissociation constants are temperature-dependent. By substituting these relations at equilibrium to the local charge neutrality principal, Eq. (17) can be rewritten as following

$$\left(1 + \frac{k_H^A}{K_A}P_{NH_3}\right)[H^+]^3 + Z_{net}[H^+]^2 - (K_W + K_N k_H^N P_{HONO} + K_{1c} k_H^C P_{CO_2})[H^+]$$
$$- 2K_{1c}K_{2c} k_H^C P_{CO_2} = 0 \tag{23}$$

where $Z_{net} \equiv 2[Ca^{2+}] + [Z^+] - [NO_3^-]$ denotes the net charge of fee cations and anions that are not constrained with gaseous compounds. The pH at equilibrium is the positive real root of the cubic polynomial. Since the partial pressures of gaseous compounds are constants in the model, concentrations of $Ca^{2+}$, $Z^+$, and $NO_3^-$ determine the pH value. The calculated analytic solutions of pH under Henry's and acid–base equilibria for varying mixing ratios of $NH_3$ and HONO are plotted in Supplementary Fig. 2. For this plot, the partial pressure of $CO_2$ was assigned as constant, 383 ppm.

**Estimation of gaseous efflux.** The gaseous efflux dynamics is obtained simply by using Henry's law and the assumption of immediate equilibration of soil air to the same partial pressures as the atmospheric level when the spatial element is connected to the atmosphere by gas phase percolation. Basically, the amount of local gaseous efflux was calculated element-wise and using the invasion percolation, the emission from atmosphere-connected clusters is summed as the gaseous efflux. When the gas percolation cluster was not connected to atmosphere directly, it stays locally as the stored gas pocket in the model and possibly increases the local concentration of dissolved gases compounds. This method has a benefit for the system like biocrusts in arid and semiarid area because; firstly, it considers the sharp gradient over the shallow depth (a few cm) which cannot be estimated by gradient method; secondly, large temperature variation during diurnal cycles and temperature-dependent-solubility of gaseous elements can be included; lastly, it includes the physical structure of unsaturated soils shaping the water and gas configuration (percolation of gas phases over the soil depth) under dynamic hydration conditions. The amount of degassed substance is solely determined by Henry's law and local mass conservation during mass transfer. The concentrations at the mass transfer equilibrium, $C^{g*}$ (gas phase) and $C^{l*}(\vec{r},t)$ (liquid phase), are determined as following

$$C^{g*}(\vec{r},t) = \begin{cases} \frac{V_l(\vec{r},t)C^l(\vec{r},t) + V_g(\vec{r},t)C^g(\vec{r},t)}{H_{cc}(T(\vec{r},t))V_l(\vec{r},t) + V_g(\vec{r},t)} & \text{if } \vec{r} \notin \Omega_a \\ p(\vec{r},t)\rho_a(T(\vec{r},t)) & \text{if } \vec{r} \in \Omega_a \end{cases} \tag{24}$$

$$C^{l*}(\vec{r},t) = \begin{cases} H_{cc}(T)C^{g*}(\vec{r},t) & \text{if } \vec{r} \notin \Omega_a \\ p(\vec{r},t)\rho_a(T(\vec{r},t))H_{cc}(T(\vec{r},t)) & \text{if } \vec{r} \in \Omega_a \end{cases} \tag{25}$$

where $V_l(\vec{r},t)$ and $V_g(\vec{r},t)$ are the volume of liquid and gas phases of the spatial element (a patch) at location $\vec{r}$ at time $t$. $\Omega_a$ is the cluster region that are connected to the atmosphere. Here, the local temperature, $T(\vec{r},t)$, determines the local solubility and air density. From the concentrations at mass transfer equilibrium, the amount of degassed/dissolved substance can be obtained locally

$$F(\vec{r},t) = C^l(\vec{r},t) - C^{l*}(\vec{r},t). \tag{26}$$

By integrating over the gas percolating region, total gaseous emission can be determined as

$$E_{tot}(t) = \int_{\Omega_a} F(\vec{r},t)d\Omega_a. \tag{27}$$

**Use of a planar optode and a microelectrode.** We used two commercial devices to measure changes in pH of drying soils: (1) a planar optode sensor for pH

(product code SF-HP5-OIW; Presens GmbH, Rosensburg, Germany) which has the precision of 0.01 pH unit at pH 7 (https://www.presens.de/products/detail/ph-sensor-foils-sf-hp5r.html), a camera (VisiSens detector unit DU02; Presens GmbH) and VisiSens software (PreSens GmbH), (2) a microelectrode with internal reference (PH-200C, Unisense, Aarhus, Denmark) with the detection limit of 0.01 pH unit ((https://www.unisense.com/pH) and SensorTracePRO software (Unisense). Before the measurements, both instruments were carefully calibrated following each instruction under lab conditions (under the ambient temperature, 22°Celcius, and under darkness).

**Sample preparation**. In this study, we used a small glass cuvette ($20 \times 20 \times 20$ mm) (See Fig. 3 and Supplementary Fig. 3) filled with (1) 2% agar saturated with phosphate-buffered saline (PBS) solution (pH = 6.1, 0.1 M) (2) sterilised quartz sand (gamma ray) with grain size in the range of 0.08~3 mm initially saturated with deionised water. For the planar optode, the sensor foil was mounted onto the bottom of the sample cuvette following the manual provided by PreSens. For the PBS solution, we mixed $NaH_2PO_4$ and $Na_2HPO_4$ based on the protocol of Sørensen's buffer. For the wet soils, gamma-ray sterilised quartz (0.08–2 mm) sand were used to avoid the effect of biological activity, and other unknown chemical processes.

**Control of the composition of gas phase**. A small glass cuvette was designed with a hole on one side, which can be used for inlet/outlet of the airflow to the sample (Supplementary Fig. 3c, d). The composition of air in the sample was controlled by injecting mixture of air and carbon dioxide ($CO_2$). For mixing the gas in situ, we used a rotameter (product code: FL-2AB-04SA; OMEGA Engineering, Manchester, UK). For airflow we maintained a constant relative humidity of 20%, a fixed rate of $1 L min^{-1}$ using a dew point generator (LI-610; LI-COR, Lincoln, USA) under a constant temperature of the lab conditions (22 °C, 40% RH). The flow rate results in a turn over rate of the air in the sample to be about $2 s^{-1}$. Supplementary Fig. 4a shows an image of the entire setup.

**Monitoring hydration conditions**. The hydration status of the sample (evaporative mass loss) was monitored simultaneously by logging the sample weight during drying (Supplementary Fig. 4a). By mapping the water loss and the water content of the sample, changes in pH are given as a function of hydration status, such as gravimetric water contents (see Figs. 3 and 4). The glass cuvette was filled with quartz sand grains up to about 3 mm. Assuming the porosity 0.4 for the sand, at least 0.5 ml of water was added to achieve a full saturation. After adding deionised water to the sample, weight of the sample and the response of planar optode were simultaneously recorded until the sample is completely dried (Supplementary Fig. 4a).

## Data availability

Source data for figures are provided with the paper. Other relevant data are available on request from the corresponding author (M.K.).

## Code availability

The MATLAB codes of the desert biocrust model (the DBM) are available on a GitHub repository at http://github.com/minsughim/DBM-for-drying-soils.

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

## Acknowledgements
We thank Daniel Breitenstein (ETHZ) for technical assistance in the lab, Daniel Baumann for IT support, Minsung Kim for the illustration, and Samuel Bickel (ETHZ) for constructive discussion. Prof. Dr. Michael Sander (ETHZ) kindly provided sterilised quartz sand and the rotameter for the measurements. Kurt Barmettler (ETHZ, Soil Chemistry group) also kindly provided a multimeter for microelectrodes. Authors acknowledge Prof. Dr. Bettina Weber (Max Planck Institute for chemistry and University of Graz) and her coworkers for helpful comments and their data on HONO emissions from drying biocrusts. We would like to thank the editor and the reviewers for their many insightful comments and suggestions that helped to improve the paper. This work was supported by European Research Council (ERC) Advanced Grant (320499-SoilLife).

## Author contributions
D.O. and M.K. conceived the research. M.K. wrote the code of the D.B.M. and performed experiments. M.K. and D.O. carried out the analysis of results and co-wrote the paper.

## Additional information

**Competing interests:** The authors declare no competing interests.

