## [Peer Review File · Nature Communications]

Reviewers' comments:

Reviewer #1 (Remarks to the Author):

The major claim in the paper 'Microscale pH variations in waters films affect HONO and NH₃ emission from drying soil and desert biocrusts' is that localized pH and microbial activity enhance HONO and NH₃ during dessication of biological crusts soils. Although this work is an important contribution to understand nitrogen gas emissions from soils, this research and the way the manuscript is framed will be of interest to a limited audience of soil scientists. The paper would be greatly strengthened by framing the work in terms of the importance of these processes in increasing HONO and NH₃ emissions. It's not entirely clear why the focus is primarily on biological soil crusts (although I think they're wonderful!) since they are important in limited parts of the globe. I do think the research will influence the thinking in the field of the controls on N gas fluxes. Again, I do think that this work is important but likely not of interest to a broader community.

Reviewer #2 (Remarks to the Author):

General comments:

The manuscript 'Microscale pH variations in water films affect HONO and NH₃ emissions from drying soils and desert biocrusts' by Kim and Or presents a modeling exercise which explores the variation of nitrous acid (HONO) and ammonia (NH₃) emissions from desert biocrusts and drying soils induced by pH variations in thin aqueous films. The modeling work is interesting and it is supported by simple experimental measurements which have the purpose to illustrate the pH zonation occurring in drying soil (using quartz sand as a surrogate). In addition, the model well reproduces previously published data of HONO from cyanobacteria-dominated crust in South-Africa.

The main claim of the paper is that the emissions of HONO and NH₃ cannot be predicted by average soil conditions and that pH zonation plays a key role in regulating such emissions. More precisely, the authors propose that the local amount of nitrate is the primary determinant of local pH during evaporative losses. The manuscript is well written, and it is interesting to read. The statistical analyses are appropriate, and the work has been performed in a rigorous way. The paper is of broad interest for the soil biogeochemistry community and provide new insights on a topic that has been only partially explored. However, the paper can further improve by adding some clarifications, such as the effect of EPS and soil pore oxygen on the presented dynamics and by providing some

additional details. For instance, why did the authors not use a sterilized soil biocrust to show the pH zonation instead of quartz sand?

Some suggestions:

1. The abiotic and biotic processes that regulate the HONO and NH₃ emissions are complex. I would suggest that the authors seek a more pedagogical way to explain their results to a broad audience. This would help the generic reader (i.e. one that is not expert in soil biocrust dynamics or modelling) to have an immediate understanding of the paper and its objectives (e.g. by adding a figure which illustrates the main finding of the paper).

2. Could the authors add a test using a real sterilized biocrust (e.g. see [1])?

3. Could the authors add some information the partitioning of N-gases in soil biocrusts? Also, given the plethora of N-gases that are emitted from biocrusts and drying soils, it would be interesting to define how the changes in HONO and NH₃ emissions affect the production/emissions of other N-gases. Can the authors add some remarks on the effect of the presented dynamics on the global N losses from biocrusts?

4. EPS has shown to play an essential role in biocrust recovery. How including EPS would change the model output? Dr. Or and others have shown the importance in wetting/drying of soil amended with Xanthan [2] or real EPS [3]. More recently the importance of EPS has been shown in biocrusts [4]. Please address this point.

5. Some consideration of soil pore oxygen can be added in the section exploring microbial activity. See [5,6].

Specific comments:

1. Title: The title reads 'Microscale pH variations in water films affect HONO and NH₃ emissions from drying soils and desert biocrusts', however the initial part of the introduction focuses solely on biocrusts and their surrounding micro-habitats. I would either change the title or rework the introduction. Soils are discussed only on page 5.

2. Line 1, page 4: What about the N-gas partitioning? Which are the predominant N-gases emitted during biocrust drying? Can the authors add some extra information about the relative importance of the different N-gases?

3. Line 2, page 7: What about the effect of EPS on the dehydration process? Why EPS are not considered in the model? Prof. Or himself and other authors have shown the importance of EPS in artificial [2] and natural [3] bio-amended soils (see [2] and [3]), as well as in biocrusts [4].
4. Line 5, page 4: Why did the authors focus their attention only on HONO and NH₃? How changes in HONO and NH₃ production/emissions will affect the other N-oxides emissions? Can the authors provide some insights?
5. Line 5, page 5: Which is the typical pH value of a soil biocrust? Are they mainly acid or alkaline?
6. Line 6 page 5: Not clear. What do the authors mean by 'similar characteristics'? How can soil texture affects saturation/desaturation and thus water content (as states at line 18) but not HONO emissions? Please clarify this point.
7. Line 18, page 5: Please add a reference.
8. Line 22 page 7: Another important factor which is not mentioned is the role oxygen. How this will affect HONO and NH₃? (See for instance [5, 6]). Can the authors incorporate some concepts into the introduction?
9. For the sake of clarity, can the authors specify if they refer to aqueous or gaseous diffusion throughout the manuscript and the figures?
10. Line 4 page 10: I would specify that these tests are done to show the abiotic effects on pH zonation at the micro-scale. Why did not the authors add a test on a real biocrust to show also the biotic effects on pH zonation instead of using only the model?
11. Line 6 page 10: How did the authors obtain the line of figure 1 for the optode? Did they compare the spatially averaged values of pH obtained with the optode to the ones obtained with the

electrode? If yes, can the authors include the standard deviation to show the spatial variability of pH values across the area of the sensor foil? How can the authors compare the spatial measurements to the punctual ones?

12. Line 4, page 12: The fact that there are no emissions at high soil moisture is not simply a direct consequence of the proposed diffusion coefficient (as showed in the supplementary information)?

13. Lines 12-13 page 10: Which is the precision of the optode with respect to the electrode? Can the authors compare the value of the pixel where they placed the sensor to the one of the microelectrode?

14. Figure 3: Could the authors include the changes in water content level with time?

15. Figure 4: Would not different soil textures have different desaturation curves?

16. Line 15, page 13: Can the author make a 3D plot adding the fluxes of HONO versus the variations in pH ?

17. Method section and/or information of the supplementary material: These sections should contain all the needed details to reproduce the experiments. Please add information on sensors calibration and the position/settings of the Visisens camera with respect to the experimental device. E.g. was the experiment conduct at constant temperature?

References:

[1] Weber, Bettina et al. (2015). "Biological soil crusts accelerate the nitrogen cycle through large 26 NO and HONO emissions in drylands". *Proceedings of the National Academy of Sciences* 112.50, 27 pp. 15384–15389.

[2] Or, Phutane, Dechesne, Ection (2007a). Extracellular polymeric substances affecting pore-scale hydrologic conditions for bacterial activity in unsaturated soils. *Vadose Zone J.* 6, 298–305.

[3] Rubol, Freixa, Carles-Brangarí Fernàndez-Garcia, Romaní, Sanchez-Vila (2014) "Connecting bacterial colonization to physical and biochemical changes in a sand box infiltration experiment", *Journal of Hydrology*, Volume 517, Pages 317-327.

[4] Chock, Antoninka, Faist, Bowker, Belnap, Barger (2018) "Responses of biological soil crusts to rehabilitation strategies", *Journal of Arid Environments*.

[5] Silver, Lugo, Keller (1999) 'Soil oxygen availability and biogeochemistry along rainfall and topographic gradients in upland wet tropical forest soils" *Biogeochemistry* 44: 301.

[6] Rubol, Manzoni, Bellin, Porporato (2013) 'Modeling soil moisture and oxygen effects on soil biogeochemical cycles including dissimilatory nitrate reduction to ammonium (DNRA)' *Advances in Water Resources*, Pages 106-12.

Response to Reviewers' Comments: NCOMMS-18-28408

Microscale pH variations during drying of soils and desert biocrusts affect HONO and NH₃ emissions

Minsu Kim^{1,2*} and Dani Or¹

¹ Soil and Terrestrial Environmental Physics (STEP),
Department of Environmental Systems Sciences (USYS), ETH Zürich,
8092 Zürich, Switzerland

² Laboratory for Air Pollution/Environmental Technology,
Empa (Swiss Federal Laboratories for Materials Science and Technology),
8600 Dübendorf, Switzerland

Correspondence to: Minsu Kim (minsukim@empa.ch)

We thank the reviewers and the editor for the many constructive comments and suggestions that helped to improve the manuscript. In the following, we provide a point-by-point response to all comments. We note that line number references are to those of the revised version of the manuscript without track changes. Appended to the response to Reviewers' comments is a copy of the original manuscript marked with all the changes made during the revision process. The new text is in blue while the crossed-out text in red refers to the deleted original text.

Reviewers' comments:

Reviewer #1 (Remarks to the Author):

The major claim in the paper 'Microscale pH variations in waters films affect HONO and NH₃ emission from drying soil and desert biocrusts' is that localized pH and microbial activity enhance HONO and NH₃ during desiccation of biological crusts soils. Although this work is an important contribution to understand nitrogen gas emissions from soils, this research and the way the manuscript is framed will be of interest to a limited audience of soil scientists. The paper would be greatly strengthened by framing the work in terms of the importance of these processes in increasing HONO and NH₃ emissions. It's not entirely clear why the focus is primarily on biological soil crusts (although I think they're

wonderful!) since they are important in limited parts of the globe. I do think the research will influence the thinking in the field of the controls on N gas fluxes. Again, I do think that this work is important but likely not of interest to a broader community.

We thank the reviewer for the encouraging comments, and the important suggestion to provide a broader context for the study and the new findings. In the revised manuscript, we followed the recommendation and incorporated the comments into introduction and discussion. We attempted to generalize findings and to discuss their implications in a wider context without straying from the evidence supported by this study. Specifically, we have broadened the discussion to include any soil surfaces in addition to the compelling experimental and modelling results specific to desert biocrusts. We hope that these changes and expanded scope will be of interest to a broad audience.

Reviewer #2 (Remarks to the Author):

General comments:

The manuscript 'Microscale pH variations in water films affect HONO and NH₃ emissions from drying soils and desert biocrusts' by Kim and Or presents a modeling exercise which explores the variation of nitrous acid (HONO) and ammonia (NH₃) emissions from desert biocrusts and drying soils induced by pH variations in thin aqueous films. The modeling work is interesting and it is supported by simple experimental measurements which have the purpose to illustrate the pH zonation occurring in drying soil (using quartz sand as a surrogate). In addition, the model well reproduces previously published data of HONO from cyanobacteria-dominated crust in South-Africa.

The main claim of the paper is that the emissions of HONO and NH₃ cannot be predicted by average soil conditions and that pH zonation plays a key role in regulating such emissions. More precisely, the authors propose that the local amount of nitrate is the primary determinant of local pH during evaporative losses. The manuscript is well written, and it is interesting to read. The statistical analyses are appropriate, and the work has been performed in a rigorous way. The paper is of broad interest for the soil biogeochemistry community and provide new insights on a topic that has been only partially explored. However, the paper can further improve by adding some clarifications, such as the effect of EPS and soil pore oxygen on the presented dynamics and by providing some additional details. For instance, why did the authors not use a sterilized soil biocrust to show the pH zonation instead of quartz sand?

We thank the reviewer for the supportive comments and the insightful observations and suggestions. Following the reviewer's suggestions, we have rewritten parts of the manuscript to strengthen the main findings and to address some of the shortcomings identified by the reviewer. In the following, we provide detailed responses to the reviewer's comments and highlight the corresponding revisions.

Some suggestions:

1. The abiotic and biotic processes that regulate the HONO and NH₃ emissions are complex. I would suggest that the authors seek a more pedagogical way to explain their results to a broad audience. This would help the generic reader (i.e. one that is not expert in soil biocrust dynamics or modelling) to have an immediate understanding of the paper and its objectives (e.g. by adding a figure which illustrates the main finding of the paper).

In response to the reviewer's suggestion, we have augmented a conceptual figure. The illustrative schematic consists of a diagram that shows how pH zonation in a drying soil supports the formation of HONO emission hotspots (Fig.1 in the revised manuscript). We hope that this newly generated figure adds clarity to the discussion.

2. Could the authors add a test using a real sterilized biocrust (e.g. see [1])?

While such a test could provide additional insights and support the proposed formation of abiotic pH zonation during drying, the experimental challenges and complexities of using natural biocrusts definitively are beyond the scope of this study. Our experimental setup was designed to provide a proof of concept. Thus, avoiding the complexities of natural biocrusts with poorly constrained composition and the potential side effects of sterilisation (e.g. survival of spores or modification of physico-chemical properties) was the primary concern. We have emphasized in the manuscript that we rely on reported experimental results for HONO emission dynamics and have addressed the raised question explicitly in the revised discussion section (page 9, lines 23-32). We want to conduct additional experiments to evaluate the model and proposed mechanisms in future studies which would include natural soils and biocrust samples.

3. Could the authors add some information the partitioning of N-gases in soil biocrusts? Also, given the plethora of N-gases that are emitted from biocrusts and drying soils, it would be interesting to define how the changes in HONO and NH₃ emissions affect the production/emissions of other N-gases. Can the authors add some remarks on the effect of the presented dynamics on the global N losses from biocrusts?

We thank the reviewer for this important comment that improved the manuscript with the more specific perspectives on other N-gas pathways. As mentioned by the reviewer the various N-gases that could potentially be emitted from biocrust makes this unique ecosystem both, interesting to study and difficult to decipher. We have embraced the more complete descript of N gases partitioning in the reported study, nevertheless, we can only speculate regarding their magnitudes and global contribution. Our model is not able to capture all the coupled nonlinear dynamics that could occur when modifying the gaseous composition (and the responses of the many actors involved). Motivated by the reviewer's suggestions, we revised the introduction by adding references to gaseous composition and reframed the discussion to put potential N-losses into a global context.

4. EPS has shown to play an essential role in biocrust recovery. How including EPS would change the model output? Dr. Or and others have shown the importance in wetting/drying of soil amended with Xanthan [2] or real EPS [3]. More recently the importance of EPS has been shown in biocrusts [4]. Please address this point.

The important aspect of EPS in biocrusts are now included in the discussion section (page 10, lines 20-27). We note that the model does not explicitly include certain aspects such as the physical swelling and shrinking dynamics or the specific increased water holding capacity that have been attributed to systems with copious amounts of EPS. However, In exploring indirectly, the role of EPS on different drying rates (Figs. 6 and 7), we attempted to address the dynamic effects of EPS including changes in the water retention during evaporation process [1]. The slower drying provides a wider time window to bacterial activity, hence resulting in larger amounts of HONO emission characterised by a broader (more spread) emission peak.

5. Some consideration of soil pore oxygen can be added in the section exploring microbial activity. See [5,6].

Together with the discussion of potential effect of EPS, we have included considerations regarding the oxygen distribution within biocrusts and their potential impact on chemical and biological processes (page 10, lines 30 – page 11, lines 2).

Specific comments:

1. Title: The title reads ‘Microscale pH variations in water films affect HONO and NH₃ emissions from drying soils and desert biocrusts’, however the initial part of the introduction focuses solely on biocrusts and their surrounding micro-habitats. I would either change the title or rework the introduction. Soils are discussed only on page 5.

Following the suggestion, we have revised the introduction to cover the general aspects of N loss mechanisms from soils beyond the previous focus on desert biocrusts. For generality of the findings (while adhering to space limitations for manuscripts considered for Nat. Comm.), we have shortened the title as **“Microscale pH variations during drying of soils and desert biocrusts affect HONO and NH₃ emissions”**.

2. Line 1, page 4: What about the N-gas partitioning? Which are the predominant N-gases emitted during biocrust drying? Can the authors add some extra information about the relative importance of the different N-gases?

We appreciate the reviewer's valuable comment that helped us to reframe the discussion and consider wider aspects of N cycling. In the revised introduction, we have included various forms and pathways for N losses as following (page 3, lines 20-24).

“Gaseous N emissions from desert environments (prominently by biocrusts) include N₂O, nitric oxide (NO), nitrous acid (HONO), and ammonia (NH₃). Nearly all possible N gases emitted from desert soils result from coupled biotic and abiotic processes. NH₃ volatilisation has been shown to be the major loss of N_r gas from deserts owing to its high alkalinity (average pH ~8).”

3. Line 2, page 7: What about the effect of EPS on the dehydration process? Why EPS are not considered in the model? Prof. Or himself and other authors have shown the importance of EPS in artificial [2] and natural [3] bio-amended soils (see [2] and [3]), as well as in biocrusts [4].

We note that important aspects of EPS production and accumulation have been included in the original model. However, as the reviewer pointed out, the effects on dehydration of EPS-rich system were not considered mechanistically. We recognize the importance of EPS as a dominant component of desert biocrusts (affecting its architecture and mechanical properties). As well as the effects of EPS on hydration properties (water retention, infiltration rates and therefore gas and nutrient diffusion) [1,2]. The omission of detailed and mechanistic dehydration dynamics of EPS is a simplification that allow us to focus on complex interdependency of other variables, such as temperature, solution pH, amount of cations, biological activity and more. However, we note (as explained in the responses above) that the model could incorporate influences of EPS by alteration of the domain geometrical and water retention properties (roughness of the surfaces, porosities, or distribution of pore sizes). As an example, the figure below illustrates sample calculations of changes in water retention curve due (indirectly) to the presence of EPS.

The higher amount of EPS in the domain would lead to increase of air-entry values, decrease of average pore sizes (pores will be filled with EPS), thus would delay the evaporation process during a course of wet-dry cycle (which is in agreement with the recent work of [1]). Considering these aspects, we reported a scenario of slow-drying instead of including ill-defined dehydration process of EPS.

We appreciate this valuable comment and input. We have mentioned several important aspects of EPS for biocrust in the revised discussion section (for example, see page 10, lines 20-27, page 11 lines 18-19). We also mentioned the model's limitation and potential regarding the EPS dehydration process in the revised version.

4. Line 5, page 4: Why did the authors focus their attention only on HONO and NH₃? How changes in HONO and NH₃ production/emissions will affect the other N-oxides emissions? Can the authors provide some insights?

We have focused on the close relation of HONO and NH₃ in terms of biotic and abiotic processes. Specifically, we considered, nitrification, which results in sinks and sources of these gases, as a biotic process, while their pH dependency is an abiotic process. Other gases, such as NO or N₂O are not considered in this work and we agree that including these two gases would affect the predicted amount of HONO and NH₃ emissions in this study. Considering that N₂O emission is dominantly driven by denitrification (anaerobic processes), emission of N₂O may be negligible for drying surface soils. However, NO emission caused by nitrification process should be a highly relevant process to improve the current desert biocrust model. Furthermore, the recent publication about NO as an obligatory nitrification intermediate [3] adds the necessity of such improvement of the current model. Including NO in the model will affect the amount of NO₂⁻, thus efflux of HONO during a desiccation can be

reconciled. Such improved model will be able to show the NO emission patterns, which are similar to HONO, and to complete the picture of N partitioning within drying soils and desert biocrusts.

In the revised manuscript, we clarified our focus on HONO and NH₃ as a motivation of this study in the introduction (page 3, lines 22-30) and added the aspect of considering other gases, especially NO, to the discussion (page 11, lines 19-23).

5. Line 5, page 5: Which is the typical pH value of a soil biocrust? Are they mainly acid or alkaline?

Biocrusts develop in arid and semiarid regions where soils are mostly alkaline. Soil pH reported in the WoSIS database [4] shows that most measurements of soil pH in deserts are distributed around a value of 8 (figure below). In this figure, we classified biomes according to Olsen et al., 2001 [5].

Biocrusts samples in a study by Weber et al. 2015 are in the range of 6.83 to 8.17. We have added this refinement in page 3 line 24

6. Line 6 page 5: Not clear. What do the authors mean by ‘similar characteristics’? How can soil texture affects saturation/desaturation and thus water content (as states at line 18) but not HONO emissions? Please clarify this point.

We have rewritten the sentence to clarify (Page 4, lines 1-2) as

“Moreover, the emission patterns of HONO in drying soil exhibit similar characteristics for different soil types and are characterized by a peak in emission at a certain, “optimal” water content.”

7. Line 18, page 5: Please add a reference.

Amended (page 4, line 12)

8. Line 22 page 7: Another important factor which is not mentioned is the role oxygen. How this will affect HONO and NH₃? (See for instance [5, 6]). Can the authors incorporate some concepts into the introduction?

We have included the aspect of oxygen and its distribution in the discussion section instead of the introduction (see page 10, lines 30 – page 11, lines 2)

9. For the sake of clarity, can the authors specify if they refer to aqueous or gaseous diffusion throughout the manuscript and the figures?

We thank the reviewer for this important comment - we have clarified the terminology regarding aqueous and gaseous diffusion throughout the manuscript and thereby removed this ambiguity.

10. Line 4 page 10: I would specify that these tests are done to show the abiotic effects on pH zonation at the micro-scale. Why did not the authors add a test on a real biocrust to show also the biotic effects on pH zonation instead of using only the model?

As we explained in our general response to suggestion 2, experiments using natural biocrusts require a dedicated and complex experimental setup that is (presently) beyond the scope of this study. The main focus was on developing the model and to provide a proof of concept. The subject of pH zonation would require many tests in different natural environments to be verified. We have limited our study and experiments to the use of sterilised quartz sand and simple soil as stated below: (page 6 lines 29-32)

“The primary objective of this simple test was to illustrate the abiotic mechanism for the onset of pH zonation during soil drying at the micro-scale. We opted for using a simple system to avoid complexities of natural soils with poorly defined composition and unconstrained microbial activity that would require dedicated experimental setups to evaluate the far more complex role of the microbial contribution to the phenomenon.”

11. Line 6 page 10: How did the authors obtain the line of figure 1 for the optode? Did they compare the spatially averaged values of pH obtained with the optode to the ones obtained with the electrode?

If yes, can the authors include the standard deviation to show the spatial variability of pH values across the area of the sensor foil? How can the authors compare the spatial measurements to the punctual ones?

Yes - we compared spatially averaged optode measurements (over an area of 25 mm²) with temporally averaged electrode measurements (measured at 5 sec. intervals and averaged each 1 min.). The tip of the electrode was in close proximity to the surface of the optode. Fig. 3 includes the spatial variance of optode measurements. However, the spatial variability was smaller than the size of the symbol in the figure but is expected to increase under dryer conditions (see Fig. 4). Spatial and temporal averages are reported to compare concurrent measurements of changes in pH using two independent methods (optode and electrode, respectively).

12. Line 4, page 12: The fact that there are no emissions at high soil moisture is not simply a direct consequence of the proposed diffusion coefficient (as showed in the supplementary information)?

The reviewer is correct in the observation that reduced emission values are indeed a result of lower gas diffusivity values under saturated conditions (as also described in the supplementary information). Nevertheless, we would like to draw attention to the fact that NH₃ emission did not completely vanish (unlike HONO emissions). Fig. 5d shows modelling results of the gaseous NH₃ and HONO emissions indicating a positive efflux of NH₃ (about 50 ng.m⁻².s⁻¹ on average) even under fully saturated condition (Fig. 5a). These emissions occur at the surface of the soil where biocrusts are in contact with the air regardless of the lower gas diffusivity within the soil. At the same time, HONO emissions remain low because biocrusts are largely alkaline (average pH above 8). This reduces protonation of nitrite and thereby the formation of nitrous acid (HNO₂) regardless of the high activity by ammonia oxidisers. The increase in NH₃ efflux occurs when gas is able to percolate the soil domain after three hours (marked as the dotted arrow on the left side in Fig.5a and d). This illustrates that diffusion of NH₃ gas is hindered under saturated conditions. In contrast, HONO emissions occur when local pH drops below the pKa value of nitrite (about 3.3) and subsequent protonation of nitrite to HNO₂ happens (hence HONO emissions follow Henry's law at equilibrium).

We thank the reviewer for this comments that helped improve the description of Fig. 5 and the aspects above were added to the discussion section (page 8, lines 7 -12)

13. Lines 12-13 page 10: Which is the precision of the optode with respect to the electrode? Can the

authors compare the value of the pixel where they placed the sensor to the one of the microelectrode?

The precision of the optode sensor (Presens) is about 0.01 pH units at pH 7 (see the link <https://www.presens.de/products/detail/ph-sensor-foils-sf-hp5r.html>). For the pH electrode (Unisense), the spatial resolution is about 100-250 micrometer (size of the tip), and the detection limit is 0.01pH units (<https://www.unisense.com/pH>). We have added the precision of the measurement to the main text (page 7 lines 3-5) and the method section (page 12, lines 23-24).

It is challenging to directly compare the measurement at the tip of the electrode to the exact location on the optode due to the opaque nature of the pH sensor and soil/quartz particles. Additionally, the electrodes are easily damaged by lateral stresses making it merely impossible to accurately control their position within the sample.

14. Figure 3: Could the authors include the changes in water content level with time?

In Fig. 3c and e, we have presented the level of hydration conditions as a function of time. In these measurements, we originally measured the hydration conditions by weight (absolute loss of water during drying) and converted the weight to the corresponding water contents. For the case of drying PBS buffer in agar, the volume of the glass cuvette was used for conversion. The equivalent height of water (h_w [mm]) was calculated by dividing by the area of the cuvette. For the case of drying quartz sand, we provided the gravimetric water content (θ_g [g.g⁻¹]), which is the fraction of water to the total mass of the saturated sample.

15. Figure 4: Would not different soil textures have different desaturation curves?

Yes - soil texture will affect the shape and parameters of the soil water retention curve as reported in various databases (e.g. UNSODA [6], Rosetta [7]). These are likely to affect the drying behaviour of surfaces as explained in a recent study by Lehmann et al. 2018 [8]. We note that the porous medium used for the measurements reported in Fig. 4 was composed of fine and coarse textures similar to natural soils consisting of a distribution of grain sizes. We reported the water content deduced from changes in weight of the sample while drying under laboratory conditions (22-24°C, 30-40%RH). The rate of drying could be modified by environmental conditions and the soil water retention curve of the porous media investigated. The results in Fig. 4 illustrate that pore and surface heterogeneity can enhance pH zonation during drying (inducing films and aqueous phase fragments). For example, the

coarse-grained sector dried faster than the fine-grained sector exhibiting earlier onset of localized acidification.

16. Line 15, page 13: Can the author make a 3D plot adding the fluxes of HONO versus the variations in pH?

We thank the reviewer for the valuable idea to visualise our results. We have plotted the hydration conditions (% water holding capacity), HONO emissions, and spatial variance of pH in the figure below. In the figure below, the spatial variance of pH is presented by colour instead of 3D plots. We plotted all replicated simulation results instead of the averaged values (n=8 for all cases).

The spatial variation of pH increases until the peak of HONO emission and decreases again together with the efflux. However, the case of high drying rates and high NH_3 input did not exhibit such behaviour, rather it shows the highest pH variance at the very dry end (in the figure a). This was because the effect of water removal was dominant over the narrow time window available for bacterial activity. Additionally, the low nitrification rate determined by the low NH_3 input, explain the discrepancy to figure c. We included the spatial variance of pH in Fig. 7 to emphasis on the relation between pH zonation and HONO emission. The figure in this letter is also included as a supplementary figure for the manuscript.

17. Method section and/or information of the supplementary material: These sections should contain

all the needed details to reproduce the experiments. Please add information on sensors calibration and the position/settings of the Visisens camera with respect to the experimental device. E.g. was the experiment conduct at constant temperature?

We added detailed information on the experimental set up to the method section and supplementary information following the suggestion. A picture of the exact setup, depicting the location of the VisiSense camera and the microelectrode, is additionally provided in the Supplementary Figure 4.

References:

[1] Weber, Bettina et al. (2015). "Biological soil crusts accelerate the nitrogen cycle through large 26 NO and HONO emissions in drylands". *Proceedings of the National Academy of Sciences* 112.50, 27 pp. 15384–15389.

[2] Or, Phutane, Dechesne, Ection (2007a). *Extracellular polymeric substances affecting pore-scale hydrologic conditions for bacterial activity in unsaturated soils. Vadose Zone J.* 6, 298–305.

[3] Rubol, Freixa, Carles-Brangarí Fernández-García, Romaní, Sanchez-Vila (2014) "Connecting bacterial colonization to physical and biochemical changes in a sand box infiltration experiment", *Journal of Hydrology*, Volume 517, Pages 317-327.

[4] Chock, Antoninka, Faist, Bowker, Belnap, Barger (2018) "Responses of biological soil crusts to rehabilitation strategies", *Journal of Arid Environments*.

[5] Silver, Lugo, Keller (1999) 'Soil oxygen availability and biogeochemistry along rainfall and topographic gradients in upland wet tropical forest soils' *Biogeochemistry* 44: 301.

[6] Rubol, Manzoni, Bellin, Porporato (2013) 'Modeling soil moisture and oxygen effects on soil biogeochemical cycles including dissimilatory nitrate reduction to ammonium (DNRA)' *Advances in Water Resources*, Pages 106-12.

We have included additional references together with the suggested list. We thank the reviewers for the valuable inputs that improved the manuscript.

Reference

[1] Adessi, Alessandra, et al. "Microbial extracellular polymeric substances improve water retention in dryland biological soil crusts". *Soil Biology and Biochemistry*, 116:67–69. (2018).

- [2] Or, Dani, et al. Extracellular polymeric substances affecting pore-scale hydrologic conditions for bacterial activity in unsaturated soils. *Vadose Zone J.* 6, 298–305. (2007a).
- [3] Caranto, Jonathan D and Kyle M Lancaster, “Nitric oxide is an obligate bacterial nitrification intermediate produced by hydroxylamine oxidoreductase”. *Proceedings of the National Academy of Sciences*, 114(31):8217– 8222. (2017).
- [4] Batjes, N. H., Ribeiro, E., van Oostrum, A., Leenaars, J., Hengl, T., and Mendes de Jesus, J. WoSIS: providing standardised soil profile data for the world, *Earth Syst. Sci. Data*, 9, 1-14, <https://doi.org/10.5194/essd-9-1-2017> (2017).
- [5] Olson, D. M., Dinerstein, E., Wikramanayake, E. D., Burgess, N. D., Powell, G. V. N., Underwood, E. C., D'Amico, J. A., Itoua, I., Strand, H. E., Morrison, J. C., Loucks, C. J., Allnutt, T. F., Ricketts, T. H., Kura, Y., Lamoreux, J. F., Wettengel, W. W., Hedao, P., Kassem, K. R. Terrestrial ecoregions of the world: a new map of life on Earth. *Bioscience* 51(11):933-938 (2001)
- [6] Nemes, Attila; Schaap, Marcel; Leij, Feike J.; Wösten, J. Henk M. UNSODA 2.0: Unsaturated Soil Hydraulic Database. Database and program for indirect methods of estimating unsaturated hydraulic properties. US Salinity Laboratory - ARS - USDA. <http://dx.doi.org/10.15482/USDA.ADC/1173246> (2015)
- [7] Schaap, M. G., Leij, F. J., and van Genuchten, M. Th., Rosetta: a computer program for estimating soil hydraulic parameters with hierarchical pedotransfer functions, *J. of Hydrol.*, 251, 163-176 (2001)
- [8] Lehmann, P., Merlin, O., Gentile, P., & Or, D. Soil texture effects on surface resistance to bare soil evaporation. *Geophysical Research Letters*, 45, 10,398– 10,405. (2018)

1 **Microscale pH variations during drying of soils and** 2 **desert biocrusts affect HONO and NH₃ emissions**

**Minsu Kim^{1,2*} and Dani Or¹**
¹Soil and Terrestrial Environmental Physics (STEP), Department of Environmental Systems Sciences (USYS), ETH
Zürich, 8092 Zürich, Switzerland
²Laboratory for Air Pollution/Environmental Technology, Empa (Swiss Federal Laboratories for Materials Science
and Technology), 8600 Dübendorf, Switzerland
*mins.kim@empa.ch
**ABSTRACT**

Microscale interactions in soil may give rise to highly localised conditions that disproportionately affect soil nitrogen transformations. We report mechanistic modelling of coupled biotic and abiotic processes during drying of soil surfaces and biocrusts. The model links localised microbial activity with pH variations within thin aqueous films that jointly enhance emissions of nitrous acid (HONO) and ammonia (NH₃) during soil drying well above what
would be predicted from mean hydration conditions and bulk soil pH. We compared model predictions with case studies in which reactive nitrogen gases fluxes from a drying biocrusts were measured. Soil and biocrust drying rates affect HONO and NH₃ emission dynamics. Additionally, we predict strong effects of atmospheric NH₃ levels on reactive nitrogen gas losses. Laboratory measurements confirm the onset of microscale pH localisation and highlight the critical role of micro-environments in the resulting biogeochemical fluxes from terrestrial ecosystems.

1 Introduction

Biological soil crusts (hereafter biocrusts) are dense cryptogamic communities developed on soil surfaces (mostly < 10 mm thick) in arid and semi-arid regions, and are estimated to cover about 12% of terrestrial surfaces¹. Biocrust communities constitute of photoautotrophs, such as cyanobacteria, algae, lichens, and mosses, and other heterotrophic microorganisms^{2,3}. Biocrusts are considered pioneers of dryland ecosystems due to their role as producers of fixed carbon and nitrogen and as exporters of these fixed nutrients to their surroundings and thus increase fertility of initially barren dryland soils and promote conditions for ecosystem evolution⁴.

A prominent characteristic of this live cover, that overlays many dryland surfaces, is its contribution to nitrogen cycling at regional and global scales. Estimates suggest that diazotrophs in biocrusts fix about 24.4 (3.1—45.6) Tg of N per year globally, representing 40% of terrestrial biological N₂ fixation¹. Nitrogen (N) is the most abundant element of Earth's atmosphere but occurs in an inert form (dinitrogen N₂) largely unavailable for common biological activity. N₂ gas is transformed into more reactive compounds (e.g. ammonium NH₄⁺, nitrate NO₃⁻, etc., collectively termed 'reactive nitrogen' N_r), that enable metabolism and growth of organisms. The transformation of N₂ to reactive nitrogen N_r occurs naturally in soils and is mediated by microorganisms. This crucial part of the nitrogen cycle entails nitrogen fixation, nitrification, and denitrification that produce various oxidation states of N_r^{5,6}. The partitioning of N affects soil microbial communities and depends on environmental factors such as soil type, organic carbon content, hydration, temperature, and pH^{7,8}. Biologically fixed or imported N_r in soils can be lost back to the atmosphere or leached to the ground by infiltrating water depending on the soil's environmental conditions. The soil nitrogen balance is important not only for soil fertility but also due to its roles in potent greenhouse gas emissions (e.g., nitrous oxide, N₂O) and local pollutant dynamics to surface and groundwater resources (e.g., NO₃⁻). Despite a vast body of research and observations, basic aspects concerning the fate of this large input of fixed N by biocrusts remains unclear. In desert soils, N_r in soils and its environmental controls remain uncertain due to the complex interplay between biotic and abiotic processes.

A prominent example of tightly coupled biotic and abiotic processes is found in desert environments of arid or semi-arid regions. Desert soils are known to have low soil N accretion rates, with only 10% of fixed N being retained⁹, with N loss occurring multiple pathways such as dissolution and transport with infiltrating in soil water, gaseous emissions, and erosional processes¹⁰.

Gaseous emissions of fixed N are considered the primary loss pathway, accounting for about 77% of total N inputs according to some estimates⁹. These thin surface crusts host dense microbial communities and account for 40-85% of the annual global terrestrial biological nitrogen fixation¹. The primary N_r loss pathway in these desert ecosystems is gaseous emissions, which account for 77% of total N inputs according to some estimates⁹. A suite of nitrogen oxides can be released as byproducts

of biological activity in biocrusts, including by nitrification¹¹⁻¹³ and denitrification¹⁴. The sources of abiotic
emissions are often chemical reactions mediated by solar radiation and soil moisture¹⁵ or by local acidity caused by
mineral substrates on soil surfaces^{16,17}. The form of emitted N gases from biocrusts include greenhouse gases and
reactive trace gases, such as nitrous oxide (N_r input by dry and wet deposition and biological fixation⁹). Gaseous
N emissions from desert environments include N_2O ^{14,18}, nitric oxide (NO)^{15,19-21}, nitrous acid (HONO)^{21,22}, and
ammonia (NH_3)^{15,20,23}. This study focuses on and emissions, both known to be affected by air-soil exchange
as driven by Nearly all possible N gases emitted from desert soils result from coupled biotic and biotic processes in
desert biocrusts.
and abiotic processes that primarily occur in biocrusts. NH_3 are important atmospheric trace gases, and their
emissions from biocrusts (and from soils in general) play a crucial role for atmospheric pollution at regional to
global scales. is the primary alkaline gas that regulates rain acidity, it also affects formation of clouds and
aerosols²⁴. volatilisation has been shown to be the major loss of N_r in deserts²⁰ owing to their high alkalinity
(average pH~8). Interestingly, biocrusts (largely alkaline with pH 6.8 to 8.2²¹) that provide fixed N to desert soils
also emit large amounts of HONO^{21,22}, known to form under acidic conditions because acid-base dissociation
constant of HONO is a daytime source of hydroxyl (\cdot) radical and nitric oxide (\cdot) that regulate the oxidative capacity of
the atmosphere. These two $pK_a = 3.3$ (Su et al. 2011, Maljanenet al 2013). This puzzle motivated our investigation
of gaseous emission mechanisms regarding these two important soil nitrogen compounds, NH_3 and HONO. These
are tightly coupled in terms of nitrification (biotic process) and share their pH dependency on emission in gaseous
from (an abiotic process) (Fig. 1).
During the biologically driven nitrification, ammonia oxidisers including bacteria and archaea (in this model,
these oxidisers are simply represented as AOB) transform the fixed inorganic N, ammonium (by (biotic) nitrification
(the sequential oxidation of NH_4^+), to to NO_3^- with nitrite (NO_2^-) where as intermediate product), and their pH
dependency on degassing (abiotic protonation of NH_3 and NO_2^- is transformed to nitrate (\cdot) by nitrite oxidising
bacteria (NOB). Biologically available for AOB depends on the input of fixed N and pH of soil water. At high pH,
can be emitted as gas (~9.3) where this volatilisation may suppress AOB activity. Furthermore, AOB release an
intermediate product of nitrification, \cdot , which has been suggested as a major source of emissions from soils²⁵⁻²⁷.
An essential step for such emissions, is the protonation of \cdot , forming \cdot . Since the acid-base dissociation constant of
is ~3.3, soils with low pH and high levels are expected to release a substantial amount of \cdot ²⁵. However, in contrast
with the expectation that emissions are promoted in acidic soils, evidence) (Fig. 1).
Evidence suggests that significant fluxes amount of HONO are can be emitted from neutral or alkaline soils
(above pH~5) and from desert biocrusts^{21,26}²⁶. This implies that general processes cause emissions of NH_3
and HONO from soils not limited to desert biocrusts. Moreover, the temporal emission patterns of HONO
emissions are similar characteristics across during soil drying exhibit similar characteristics for different soil
types ~~and cyanobacteria-dominated biocrusts, exhibiting emissions with a well-defined peak~~, showing a peak
emission at a certain “optimal” water content under unsaturated conditions. Studies have ~~proposed that AOB~~
~~activity could be suggested that ammonia oxidisers are~~ responsible for such distinct ~~pattern emission patterns~~ of
HONO ~~emissions from soils~~²⁶. Scharko et al. (2015)²⁷ combined flux chamber measurements with genomic ap-
proaches ~~to conclude and concluded~~ that HONO emissions were related to the abundance of ammonia oxidisers
within neutral or alkaline soils ~~(that exhibit high nitrification rates). Their genomic analysis has also shown the~~. Yet,
their genomic analysis also indicated presence of active ~~NOB that are supposed nitrite oxidisers that are expected~~
to complete the nitrification process.
~~These consistent~~ These observations raise several questions: First, the ~~observations of~~ simultaneous HONO
and NH₃ emissions from a soil or ~~bioerust appears a biocrust appear~~ to be in contradiction with the high levels of
NH₄⁺ and NO₂⁻ and bulk soil pH in equilibrium. Second, ~~if NOB are active in a soil, the~~ the presence of nitrite
oxidisers and the production of NO₂⁻ by ~~AOB as the ammonia oxidisers (a~~ direct source for HONO ~~emission~~
~~emissions)~~ must be reconciled due to the expectation of NO₂⁻ consumption by ~~NOB nitrite oxidisers~~. Finally,
a ~~characteristic prominent~~ feature of HONO emissions ~~in drying soils (and biocrusts) is the strong from various~~
~~soils while drying, points to a strong soil~~ moisture dependency irrespective of nitrifiers’ activity. ~~The~~²⁶. This
dependency on soil hydration ~~conditions motivated us to have a closer look at how changes in soil chemistry caused~~
~~by hydration dynamics affect microbial activity? and how soil pH is affected by surface drying? state motivated~~
our interest in quantifying biotic and abiotic conditions in soil during drying. How could soil pH be affected by
drying? How does microbial activity affect the aqueous phase chemistry of drying soils?
~~To address these questions we focused on soil hydration dynamics at the microbial scale. Surprisingly, effect~~
~~of hydration dynamics~~ The effects of hydration dynamics on chemical and biological process at the microscale
~~have been largely ignored although it is a ubiquitous process are poorly understood despite their ubiquity and~~
potential importance for biogeochemical processes in surface soils. ~~We~~ In this study, we employ a mechanistic
model ~~for the that integrates~~ interactions between soil properties, microbial activity, and ~~physicochemical processes~~
~~across water-air interfaces within drying soils~~ abiotic processes across air-water interfaces. We focus on the roles
of hydration dynamics and the spatial heterogeneity of soil surfaces in modifying ~~pH-related gaseous emissions.~~
local pH related to gaseous emissions, especially HONO and NH₃ (Fig. 1). We first address ~~biotic-abiotic general~~
processes occurring within drying soils and ~~then expand the picture to thin desert biocrusts.~~ demonstrate how the
microscopic hydration conditions dictate the time scales of physicochemical processes that result in localisation of
pH during drying. We then turn our focus to desert biocrusts that provide a case study of real soils and show how
nitrifiers act as sinks and sources for modifying local conditions that can cause strong variation of pH within drying
soils. The discussion follows with implications of our findings that highlight the general importance of hydration
dynamics in determining gaseous emission of N_r, relevant for global N cycling.
2 Results and Discussion
2.1 Soil hydration represented by water Water contents and water film thickness distributions
~~The~~ A quantitative description of soil gaseous ~~exchanges~~ exchange is strongly dependent on the representation of the
soil aqueous phase and air-water interfaces. Macroscopically, soil hydration is ~~characterised by water contents and~~
~~matric potentials, these interdependent variables modifying gas diffusivity and often~~ characterised by the water
content and the matric potential, both are interdependent variables that modify gas diffusivity, aqueous phase
connectivity and biological activity and thus gaseous fluxes from soil. ~~However, the macroscopic representation does~~
~~not provide~~ The macroscopic representation, however, does not represent resolved geometrical information on ~~the~~
~~distribution of~~ soil aqueous phase ~~that is shaped by complex pores and surfaces~~ at scales relevant to microbial life
(submillimetre scales)^{28,29}. ~~In this study, we use~~ (a schematic of aqueous phase distribution in various hydration
states is given in Fig. 2a). We thus employ a variable related to the water film thickness retained by rough soil sur-
faces to represent soil hydration status at the microscale ~~(and~~ as the primary interface for gas uptake and emissions).
The volume of the ~~liquid film~~ local water film also controls local concentrations of dissolved substances, thereby
determining rates and amounts of matter exchange between gas and ~~bare~~ mineral surfaces.
We implemented a previously developed rough surface model^{30–32} that links macroscopic soil water content to
microscopic aqueous film thickness at different matric potential values. ~~We define a physical domain representing~~
~~a vertical cross-section of a desert bio crust comprised of soil grain surfaces (rough solid patches) each retaining~~
~~water based on own roughness and ambient matric potential. The effective~~ (Fig. 2b). The film thickness reflects
the amount of water retained to soil grain surfaces owing to the combined effects of adsorption and capillarity
encapsulated in the definition of soil matric potential (energy state of soil water). The spatial heterogeneity of ~~pores~~
pore sizes and surface roughness yields a distribution of water film thickness across a soil domain that contributes to
the macroscopic water content (for model details see Supplementary information). ~~The model~~ Kim and Or 2016,
2017^{30,32}). Fig. 2b shows that, as ~~the~~ soil water content varies from about $0.3 \text{ m}^3 \cdot \text{m}^{-3}$ (~~total soil~~ porosity) to about
$0.01 \text{ m}^3 \cdot \text{m}^{-3}$ (residual water content) during desiccation, the effective water film thickness (per unit soil surface area)
varies by orders of magnitude from about 10^{-5} m at saturation to about 10^{-8} m (Fig. 2a,b in agreement with Tuller
and Or 2005³³). ~~Even under moderately dry conditions, a thin water film on soil surfaces serves as the gas-liquid~~
~~interface~~ This implies that the water loss at microscale cannot be scaled as the changes in water contents during
desiccation.
2.2 Time scales of physicochemical processes in unsaturated soils
Changes in the distribution of aqueous film thickness during soil desiccation affect the time scales of various processes
(Fig. 2bc). Here we focus primarily on physical and chemical processes within and across the ~~gas-liquid~~ gas-liquid
interface. Near saturation, ~~(before gas percolation, marked as a vertical dotted line in Fig. 2c),~~ water fills the soil
pores and ~~hinder gas percolation and exchange~~, whereas hinders gas exchange within the domain and nutrient
diffusion and chemical processes ~~become are~~ similar to aquatic systems. However, during soil desiccation, ~~the~~ air
percolates through empty soil pores and ~~facilitate facilitates~~ exchange of gaseous compounds to and from the residual
water film on the rough soil grains. The large surface-interfacial area of the thin water film in the soil matrix allows
instant equilibration of mass transfer; thus, dissolved gases follow Henry's equilibria. ~~Diffusivity of other compounds~~
~~through the aqueous phase~~ Meanwhile, the aqueous diffusion becomes reduced under unsaturated conditions owing
to lower connectivity and higher tortuosity of liquid phase³⁴. ~~Chemical processes, such as acid-base dissociation~~
~~or hydrolysis, are relatively fast compared with other processes. Under moderately dry conditions, the water film~~
~~is sufficiently thick to permit high water activity and dissociation processes are assumed to instantly equilibrate.~~
~~Consequently, Thus,~~ lateral solute diffusion through the ~~water film becomes~~ thin water film may become limiting
relative to gaseous exchanges in unsaturated soils. ~~In Fig. 2b, the timescale of diffusion in the aqueous phase is~~
The timescales of aqueous diffusion via thin films are estimated from $t \sim 2l/D_{\text{eff}}$ where l is characteristic diffusion
distance (~~we use here~~ $50\mu\text{m}$ ~~as~~ a representative local scale considering average inter-cell distances in soil is in
the order of 10^{-5} m ³⁵) and D_{eff} is the effective diffusivity of a solute at the given matric potential. ~~This suggests~~
~~that the productions and/or consumptions~~ (Fig. 2c). ~~Other chemical processes, such as acid-base dissociation or~~
~~hydrolysis in water films are relatively fast and are assumed to instantly equilibrate in the model. This implies that~~
~~the aqueous diffusion becomes the most limiting step in terms of abiotic processes. Thus, this renders production~~
~~and consumption~~ of dissolved compounds ~~would be that are highly~~ localised under unsaturated conditions ~~because~~
~~of the slow diffusion. Hence, distribution of soil minerals and biological entities become decisive and yield strong~~
~~spatial heterogeneity and gives rise to potential spatial heterogeneity~~ in chemical conditions.
**2.3 Mean soil pH vs. local-microscale aqueous film pH**
~~Soil pH is considered a primary attribute for soil microbial activity and community structure~~^{36,37}. Additionally,
~~soil pH has been used to describe the chemical dissociation for estimating pH-dependent gas emissions~~²⁵. However,
~~local variations in pH and spatial heterogeneity in chemical status of aqueous films under unsaturated conditions~~
~~would greatly affect microbial processes especially in dense desert bioerusts. While the soil or bioerust are~~
~~drying, the resulting changes in the~~ Changes in the aqueous phase configuration (i.e., film thickness ~~in this study~~)
distribution in drying soils and gas phase percolation jointly shape concentrations of dissolved gaseous compounds ~~as~~
, which are determined by mixing ratios of inorganic carbon and nitrogen (i.e. CO_2 , NH_3 , HONO etc.) based
on Henry's law at local scale. ~~The pH distribution under unsaturated conditions can be deduced from~~ Using these
physical conditions, we calculated the local pH distribution of unsaturated soils by assuming acid-base equilibria
and local charge balance (See Supplementary ~~information~~ Methods 1 for details). ~~The Air-soil exchange and limited~~
aqueous diffusion determine the spatial heterogeneity of pH within ~~drying soils is affected by air-soil exchange~~
~~and diffusion without considering biological activity~~ a drying soil even under the absence of biological activities.
Additionally, ~~the~~ distribution of soil minerals, such as iron, aluminium (hydr)oxides or calcite, ~~would also~~ contribute
to spatial heterogeneity ¹⁶~~and the resulting soil pH~~ of aqueous film pH at microscale ¹⁶. We note that the reactivity
of these minerals is also affected by hydration dynamics that determines ~~the~~ dissolution of gaseous compounds
(mainly CO₂). In the model, we consider a finite ~~amount of~~ exchangeable Ca²⁺ ~~is included~~ as a representative (calcite
forming) mineral to mimic calcareous desert soils ~~where most of bioerusts develop~~. Ca²⁺ precipitation ~~regulates~~
~~regulate~~ the upper bound of local pH where a finite buffering capacity could be easily exceeded in shrinking aqueous
volumes ~~of locally isolated patch of water film~~ during soil drying.
~~An additional~~ ~~A potential~~ source of spatial ~~variation in~~ ~~variations in local~~ pH is the distribution of chemical
ions in aqueous phase, such as the highly soluble NO₃⁻, that may be independent of gas phase constraints and
~~strongly affects~~ ~~could strongly affect~~ local pH. We ~~suggest~~ ~~propose~~ that non-uniform ~~distribution~~ ~~distributions~~ of
sources and sinks ~~and its limited diffusion causes~~ ~~coupled with limited lateral diffusion in aqueous films may give~~
~~rise to~~ local imbalance in free cations and anions. ~~This, thus,~~ ~~Consequently,~~ ~~this affects local pH and~~ results in
strong ~~spatial~~ heterogeneity of pH (under unsaturated conditions ~~that cannot be captured~~) ~~that would be difficult to~~
~~reconcile~~ with bulk soil pH (~~see Supplementary information~~ ~~measurements~~ (for details, see Supplementary Method
~~1, Supplementary Figure 1 and 2~~).
2.4 Spatially resolved pH measurements of drying soils
Evaporative water loss in soils increases concentrations of chemical compounds and precipitation of salts. These
changes influence acid-base dissociations that are kinetically rapid and highly ~~depending~~ ~~depend~~ on pH of aqueous
solutions. For systems with limited buffering capacity, pH is likely to vary during soil desiccation. Surprisingly, such
a local and dynamic aspect has been missing in studies that often consider a constant bulk soil pH value irrespective
of hydration conditions.
To examine the dynamic and local nature of soil pH during drying, we conducted a series of proof of concept
tests by measuring the pH of buffer solutions and wet quartz sand (sterilised) under two wet-dry cycles (~~Fig. 3~~).
~~The pH values and map~~ see Methods and Supplementary Methods 2). The primary objective of this simple test
~~was to illustrate the abiotic mechanism for the onset of pH zonation during soil drying at the microscale. We opted~~
~~for using a simple system to avoid complexities of natural soils with poorly defined composition and unconstrained~~
~~microbial activity that would require dedicated experimental setups to evaluate the far more complex role of the~~
~~microbial component of the phenomenon. The pH values and maps~~ were obtained from planar pH optodes (³⁸;
~~PreSens~~ ~~Bossfeld et al. 2010; PreSens GmbH, Rosensburg, Germany~~) and simultaneously verified using independent
~~microelectrodes~~ (³⁹; ~~Unisense~~). ~~Optode measurements showed~~ ~~measurements with microelectrodes~~ (PH-200C,
~~Unisense, Aarhus, Denmark~~) (Fig. 3a, Supplementary Figure 4b, c). ~~Optode measurements exhibited~~ a consistent
(albeit mild) decrease in pH (Fig. 3b, d magenta and purple lines, about 0.2-0.3 units with optode precision of 0.01
pH unit at pH 7) during drying confirming that the evaporative water removal that lends support to the hypothesis
that evaporation alters the pH in the remaining water films. The microelectrode (changes in hydration conditions
are given in Fig. 3c, e). The pH electrodes revealed a more drastic drop of pH of about 1 pH unit. This could
indicate (Fig. 3b, d, turquoise and orange lines). This suggests that the buffering capacities of the solution and
that of sand pore water was pore water and the solution were exceeded in the small volume of remaining water
film remaining small volumes of aqueous films. The differences in the magnitude of pH values measured by the
optodes and the electrodes may also reflect on the nature of the measurement itself (highly localised with the
electrodes and more diffused-diffusive with the optodes).
The optodes not only allowed for observations to dry conditions (drier than possible with the electrodes), but
they also provided a spatial distribution of pH values. We have used different textures of sands and modified the
levels of sand of different textures (different surface areas and retained water films) and modified pCO₂ levels in
the air injected air into the measurement cuvette (Fig. 4).
In these measurements, the sample of sterilised quartz sand was deliberately laid out with formed two distinctive
regions with fine and coarse textures to highlight grain sizes to accentuate the non-uniform pH dynamics during
drying. This-The nested behaviour in pH decrease in spatially averaged pH of the entire region highlighted relations
between local pH and soil texture local soil textures. This relation persisted under different pCO₂ levels in the
air suggesting a potential role of soil microscale structure affecting local pH dynamics (as also demonstrated by
the vertical gradient of pH during drying in Supplementary information; Supplementary Figure 5). Furthermore,
increasing pCO₂ levels increased the concentrations of carbonic acid and lowered the pH of the entire domain :
(Fig. 4c).
These results, should be interpreted with caution because the responses of the optode and electrodes were
designed primarily for wet conditions, hence we trust results from intermediate hydration conditions where the
optode remains fully hydrated (while film diffusion becomes limited). These limitations notwithstanding, these
preliminary measurements demonstrate how local pH varies during soil desiccation.
2.5 Predicting emissions dynamics HONO and NH₃ emission from drying biocrusts
We now expand the discussion
The discussion is extended from drying sterile soil to the surface layer hosting a bioerust with interacting
bacterial communities by employing a mechanistic bioerust model³² to gain insights into pH-dependent gas
emissions from bioerusts. For comparisons of model predictions with measurements a soil system with bacterial
communities. There is no doubt in microbial activities in soils would intensify the spatial heterogeneity of localised
film pH in drying soils. This is because they are the active sources and sinks of various substances including
gaseous compounds like CO₂. In this work, we show that the microbial activity within soil causes the pH zonation
during drying that would trigger the concurrent emission of HONO and NH₃. For this, we have employed a
previously developed mechanistic model of biocrust³², which describes the activity of an established microbial
communities, including nitrifiers, ammonia oxidisers (here, noted AOB) and nitrite oxidisers (NOB) together with
other members such as phototrophs, heterotrophs, and denitrifiers. For comparison of model predictions with
laboratory measurements using real biocrusts^{22,40}, we considered a wetting-drying event applied to model biocrust
under darkness (hence no photosynthesis) mimicking conditions of reported lab experiments^{21,22,40}.
Fig. 5 depicts ~~summarises the~~ simulated dynamics of drying biocrusts. During the 24 hours of simulated drying
~~(Fig. 5a), the net biogenic production rate of soil NO₂⁻ was negative ~~during drying~~ due to the consumption rates~~
~~by NOB exceeding production rates by AOB (Fig. 5b). Consequently, microbial activity (combining AOB and~~
~~NOB) ~~together,~~ did not provide a direct source for ~~emissions~~ HONO emissions in this case (the system acted as~~
~~a sink of ~~HONO~~ HONO via Henry's law). The strong variations in local pH resulted from the joint effects of~~
~~microbial activity and desiccation (Fig. 5c). Under wet conditions (high saturation), most of the domain is alkaline~~
~~(and the bulk ~~soil~~ pH is near 7), thus high levels of NH₃ volatilisation occurred at the soil surface (marked by a~~
~~positive NH₃ flux in Fig. 5d). The emission of NH₃ increased following desaturation and invasion of gas phase~~
~~through the ~~bioerust~~ domain (marked by gas percolation degree in Fig. 5a, and dotted arrow on the left side). These~~
~~reflect an impediment to gas emissions under high saturation irrespective of local chemical conditions. Furthermore,~~
~~simulations show a decrease in aqueous film pH during drying similar to observations (~~Figures~~ Fig. 3 and 4). The~~
~~resulting spatial variations in local pH span a range of pKa values for HONO with an increase in emission rates~~
~~(Fig. 5c, d, f). The local acidification of the water film drives ~~the~~ HONO release and NH₃ absorption. Following the~~
~~complete desiccation of the biocrust with the cessation of biological activity and high local acidification, HONO~~
~~efflux proceeds abiotically as outgassing by Henry's law and volatilisation (Fig. 5d green line).~~
We attribute this local acidification ~~during drying~~ to nitrification that results in accumulation of NO₃⁻ while
water is removed by evaporation (Fig. 5e, f). To examine ~~these~~ effects of hydration conditions and local nitrate
accumulation on aqueous film pH, we systematically calculated local pH as a function of nitrate amounts and matric
potentials (Fig. 5g). ~~In~~ For this calculation, we ~~ignore diffusion within the film and ignored diffusion~~ across aqueous
patches and consider evaporative concentrations and instantaneous equilibration of gas-liquid partitioning at local
scale only (the size of a connected liquid patch is of the order of 100 μm²). ~~Result suggests~~ Results suggest that the
local amount of NO₃⁻ is the primary determinant of local pH during evaporative water loss ~~. While other inorganic~~
~~because other inorganic components~~ (carbon and nitrogen ~~components~~) are constrained by ~~their~~ the protonated forms
of their gaseous compounds (e.g. NH₃ + H⁺ ⇌ NH₄⁺, HONO ⇌ NO₂⁻ + H⁺, etc.); ~~remains in the water film~~
~~due to its high solubility in water (in the range of ~10-1000 g/L) and it can be protonated to nitric acid (-) only under~~
~~in extremely acidic conditions (~-1.4). For moderately dry conditions on the soil surface (in the order of kPa), the~~
amount of nitrate is an important variable in determining local pH. This implies that the localised sources or sinks
of within unsaturated soils under limited diffusion can provide strong heterogeneity in pH covering the values for
and . Interestingly, the emitted amounts of from soils are shown to be strongly correlated with high nitrification
rate²⁷ or contents of and²², which however was not observed by²¹. This could support our hypothesis of local
acidification caused by accumulation.
**2.6 Characteristics of HONO and NH₃ emissions under different desiccation rates and atmospheric 7 ammonia levels**
Measuring local pH heterogeneity under unsaturated conditions microscale soil pH heterogeneity and separating
abiotic and biotic effects experimentally under unsaturated conditions remain a challenge. We thus use the model
to systematically evaluate HONO emissions under a range of conditions including different drying rates and
atmospheric NH₃ levels.
Desiccation rates regulate the optimal time window for HONO and NH₃ emissions (Fig. 6a, b, c, dotted
lines for slow drying and solid lines for fast drying, Supplementary Figure 6a,b,c) through their joint dependency
on water contents and pH (Fig. 7 and Supplementary Figure 7). Simulations suggest the NH₃ emissions to
occur before HONO emissions during a course of drying. Additionally, the absorption of NH₃ to water film can
be expected at is expected at the peak of HONO emissions emission illustrating the interrelation between these
two gases that are mediated by local pH in the aqueous phase. The mixing ratios of these gases in the air also
affect magnitudes of HONO emission and NH₃ absorption during drying (Supplementary information). Increasing
. Higher NH₃ levels increases the maximum emission flux of HONO by promoting AOB activity with higher
nitrification rates (See Fig. 6 and Supplementary information). Figure 6d,e,f.

[revised manuscript text omitted]

30 So far, we have shown the pH zonation in shrinking water films during drying acts as a trigger for HONO
31 and ~~these can be related to the population size and activity of diazotrophs and nitrifiers inhabiting the bioerust.~~
32 NH_3 emissions. The model of desert biocrusts enabled us to explore underlying mechanisms due to the explicit
33 representation of the microbial community and the distribution of their functional members ³². Although tested

on biocrusts, we argue that the pH zonation mechanism for HONO emission is generally applicable to any soils
since it is caused by orchestrated activities of ubiquitous nitrifiers and abiotic processes under evaporative forcing.
We presented measurements of local pH on sterilised sand as a proof of concept that lays the ground for further
experimentation using real soils with intact microbial communities. The mechanistic model was instrumental
in elucidating the puzzle of concurrent gaseous emissions (HONO and NH₃), yet various aspects of the model
can be developed further such as realistic representation of all aspects of EPS (hydration to diffusion effects).
An interesting and high priority addition would be the inclusion of a recently discovered pathway using NO as
an obligatory nitrification intermediate ⁴⁸, considering such pathway could shed light on similarity of emission
patterns of NO and HONO from drying soils. Furthermore, it would help quantifying abiotic NO emissions during
drying that could affect the activity of NOB thus modify nitrification rates and accumulation of NO₃⁻. A natural
extension of this study is to consider agricultural soils and support recent findings of anaerobic nitrate reduction
in oxygen-limited microsites that act as a source of HONO under wet conditions ⁴⁵. Such model refinements
would enhance our understanding of general mechanisms dictated by microscale processes with respect to the
factors shaping them, as shown for pH zonation driven by dynamics of soil hydration. Ultimately, this could lead
to improved predictions of nitrogen partitioning between soils and the atmosphere; a highly relevant aspect for
regional and global models of the nitrogen cycle.
**3 Methods and Materials**
**3.1 ~~The desert biocrust mathematical~~ Mathematical model of desert biocrusts**
The desert biocrust model (DBM)³² is a mechanistic model that links the aqueous state with geochemical processes
and biological activity in pioneer desert biocrusts (no lichens and mosses). The DBM considers diffusion-reaction,
mass transfer at gas/liquid interface, and chemical processes like C and N dissociation, volatilisation, and precip-
itation, whereas microbial processes are described by an individual based representation of cells. The biocrust
microbial community consists of four functional groups; photoautotrophs, aerobic heterotrophs, denitrifiers (anaero-
bic heterotrophs), and chemoautotrophs (nitrifiers; AOB and NOB). The cycles of carbon and nitrogen are performed
only by microorganisms (no higher organisms) and thus representing cyanobacteria dominated biocrusts. For
fully saturated biocrusts, the model has been tested extensively and found to agree with multiple lab experiments
in terms of dynamics of oxygen and pH profile, and CO₂ efflux from biocrust under day-night cycles³². This
study extends the previous work by exposing the microbial community to dynamic hydration conditions (wet-dry).
In other words, we have used the distribution and abundances of microorganisms obtained at full saturation as
initial conditions for the subsequent desiccation and rewetting cycles. We note that the simulations mimicked
the 'darkness' of the lab conditions, where HONO emission dynamics were measured ^{21,22,40}. Therefore, there
was no photosynthesis during drying and the activities of chemoautotrophs as nitrifiers governed the gas emission
dynamics. In this study, the atmospheric level of HONO was kept constant as 1 ppb in agreement with field measure-
ments for semiarid pine forest²⁵. The mixing ratio of NH₃ was used as a control parameter for the simulations of
FigFigs. 6 and Supplementary information. We 7. In Supplementary Figure 6, we varied the atmospheric level of
NH₃ from 0.1 ppb to ~~20~~10 ppb (representing typical values that are in the range of 1 to 10 ppb depending on the time
of the day, season, and regions). Detailed description is provided in Kim and Or (2017)~~and in the Supplementary~~
~~information~~³² and in Supplementary Method 2 for this study.
~~Detailed descriptions are provided in³² and Supplementary information.~~
**3.2 Experimental setup for localised pH**
We have used a planar pH optode sensor with the precision of 0.01 at pH 7 (PreSens GmbH, Rosensburg, Germany)
and a PH-200C microelectrode with the precision of 0.01 pH unit (Unisense, Aarhus, Denmark) that were installed
in a cubic glass sample holder (~~20 × 20 × 20 mm~~^{2x 2x 2 cm}). The cubic sample holder (Fig. 3) was filled with (1)
2% agar saturated with phosphate buffered saline (PBS) solution (pH = 6.1, 0.1 ~~M~~ ^{0M}) (2) sterilised quartz sand
(gamma ray) with grain size in the range of 0.08~3 mm initially saturated with deionised water. The sample holder
was equipped with an inlet for supply of constant gas flow to the sample. The composition of air in the sample was
controlled by injecting mixture of air and carbon dioxide (CO₂). For mixing the gas in situ, we used a rotameter
(product code: FL-2AB-04SA; OMEGA Engineering, Manchester, UK). For airflow we maintained a constant
relative humidity of 20%~~and~~, a fixed rate of 1 L.min⁻¹ using a dew point generator (LI-610; LI-COR, Lincoln, USA)
under a constant temperature of the lab conditions (22°C, 40% RH). The hydration status of the sample (evaporative
mass loss) was monitored by logging the sample weight during drying. For details of the experimental procedures
and additional measurements, see Supplementary ~~information~~^{Method 2}.
**References**
- **1.** Rodriguez-Caballero, E. *et al.* Dryland photoautotrophic soil surface communities endangered by global change.
*Nature Geoscience* **11**, 185 (2018).
- **2.** Belnap, J. & Lange, O. L. (eds.) *Biological Soil Crusts: Structure, Function, and Management* (Springer, 2003).
- **3.** Weber, B., Büdel, B. & Belnap, J. (eds.) *Biological Soil Crusts: An Organizing Principle in Drylands* (Springer,
2016).
- **4.** Pointing, S. B. & Belnap, J. Microbial colonization and controls in dryland systems. *Nature Reviews Microbiol-*
*ogy* (2012).
- **5.** Kuypers, M. M., Marchant, H. K. & Kartal, B. The microbial nitrogen-cycling network. *Nature Reviews*
*Microbiology* **16**, 263 (2018).
- **6.** Stein, L. Y. & Klotz, M. G. The nitrogen cycle. *Current Biology* **26**, R94–R98 (2016).
- **7.** Or, D., Smets, B., Wraith, J., Dechesne, A. & Friedman, S. Physical constraints affecting bacterial habitats and
activity in unsaturated porous media—a review. *Advances in Water Resources* **30**, 1505–1527 (2007).
- **8.** Tecon, R. & Or, D. Biophysical processes supporting the diversity of microbial life in soil. *FEMS microbiology*
*reviews* **41**, 599–623 (2017).
- **9.** Peterjohn, W. T. & Schlesinger, W. H. Nitrogen loss from deserts in the southwestern united states. *Biogeo-*
*chemistry* **10**, 67–79 (1990).
- **10.** Barger, N. N., Weber, B., Garcia-Pichel, F., Zaady, E. & Belnap, J. Patterns and controls on nitrogen cycling
of biological soil crusts. In *Biological Soil Crusts: An Organizing Principle in Drylands*, 257–285 (Springer,
2016).
- **11.** Johnson, S. L., Budinoff, C. R., Belnap, J. & Garcia-Pichel, F. Relevance of ammonium oxidation within
biological soil crust communities. *Environmental Microbiology* **7**, 1–12 (2005).
- **12.** Johnson, S. L., Neuer, S. & Garcia-Pichel, F. Export of nitrogenous compounds due to incomplete cycling
within biological soil crusts of arid lands. *Environmental Microbiology* **9**, 680–689 (2007).
- **13.** Strauss, S. L., Day, T. A. & Garcia-Pichel, F. Nitrogen cycling in desert biological soil crusts across biogeo-
graphic regions in the Southwestern United States. *Biogeochemistry* **108**, 171–182 (2012).
- **14.** Abed, R. M., Lam, P., De Beer, D. & Stief, P. High rates of denitrification and nitrous oxide emission in arid
biological soil crusts from the Sultanate of Oman. *The ISME Journal* **7**, 1862–1875 (2013).
- **15.** McCalley, C. K. & Sparks, J. P. Abiotic gas formation drives nitrogen loss from a desert ecosystem. *Science*
**326**, 837–840 (2009).
- **16.** Donaldson, M. A., Bish, D. L. & Raff, J. D. Soil surface acidity plays a determining role in the atmospheric-
terrestrial exchange of nitrous acid. *Proceedings of the National Academy of Sciences* **111**, 18472–18477
(2014).
- **17.** Kebede, M. A., Bish, D. L., Losovyj, Y., Engelhard, M. H. & Raff, J. D. The role of iron-bearing minerals in
NO_2 to NO conversion on soil surfaces. *Environmental Science & Technology* **50**, 8649–8660 (2016).
- **18.** Lenhart, K. *et al.* Nitrous oxide and methane emissions from cryptogamic covers. *Global change biology* **21**,
3889–3900 (2015).
- **19.** Barger, N. N., Belnap, J., Ojima, D. S. & Mosier, A. NO gas loss from biologically crusted soils in Canyonlands
National Park, Utah. *Biogeochemistry* **75**, 373–391 (2005).
- **20.** McCalley, C. K. & Sparks, J. P. Controls over nitric oxide and ammonia emissions from Mojave Desert soils.
*Oecologia* **156**, 871–881 (2008).
- **21.** Weber, B. *et al.* Biological soil crusts accelerate the nitrogen cycle through large NO and HONO emissions in
drylands. *Proceedings of the National Academy of Sciences* **112**, 15384–15389 (2015).
- **22.** Meusel, H. *et al.* Emission of nitrous acid from soil and biological soil crusts represents an important source of
hono in the remote atmosphere in cyprus. *Atmospheric Chemistry and Physics* **18**, 799–813 (2018).
- **23.** Barger, N. N. *Biogeochemical Cycling and N Dynamics of Biological Soil Crusts in Semi-arid Ecosystem.* Ph.D.
thesis, Colorado State University (2003).
- **24.** Behera, S. N., Sharma, M., Aneja, V. P. & Balasubramanian, R. Ammonia in the atmosphere: a review on
emission sources, atmospheric chemistry and deposition on terrestrial bodies. *Environmental Science and*
*Pollution Research* **20**, 8092–8131 (2013).
- **25.** Su, H. *et al.* Soil nitrite as a source of atmospheric HONO and OH radicals. *Science* **333**, 1616–1618 (2011).
- **26.** Oswald, R. *et al.* HONO emissions from soil bacteria as a major source of atmospheric reactive nitrogen.
*Science* **341**, 1233–1235 (2013).
- **27.** Scharko, N. K. *et al.* Combined flux chamber and genomics approach links nitrous acid emissions to ammonia
oxidizing bacteria and archaea in urban and agricultural soil. *Environmental Science & Technology* **49**, 13825–
13834 (2015).
- **28.** Grundmann, G. *et al.* Spatial modeling of nitrifier microhabitats in soil. *Soil Science Society of America Journal*
**65**, 1709–1716 (2001).
- **29.** Nunan, N., Wu, K., Young, I., Crawford, J. & Ritz, K. Spatial distribution of bacterial communities and their
relationships with the micro-architecture of soil. *FEMS Microbiology Ecology* **44**, 203–215 (2003).
- **30.** Kim, M. & Or, D. Individual-based model of microbial life on hydrated rough soil surfaces. *PLoS ONE* **11**,
e0147394 (2016).
- **31.** Šťovíček, A., Kim, M., Or, D. & Gillor, O. Microbial community response to hydration-desiccation cycles in
desert soil. *Scientific Reports* **7** (2017).
- **32.** Kim, M. & Or, D. Hydration status and diurnal trophic interactions shape microbial community function in
desert biocrusts. *Biogeosciences* **14**, 5403–5424 (2017).
- **33.** Tuller, M. & Or, D. Water films and scaling of soil characteristic curves at low water contents. *Water Resources*
*Research* **41** (2005).
- **34.** Moldrup, P. *et al.* Modeling diffusion and reaction in soils: X. a unifying model for solute and gas diffusivity in
unsaturated soil. *Soil Science* **168**, 321–337 (2003).
- **35.** Raynaud, X. & Nunan, N. Spatial ecology of bacteria at the microscale in soil. *PloS one* **9** (2014).
- **36.** Fierer, N. & Jackson, R. B. The diversity and biogeography of soil bacterial communities. *Proceedings of the*
*National Academy of Sciences* **103**, 626–631 (2006).
- **37.** Lauber, C. L., Hamady, M., Knight, R. & Fierer, N. Pyrosequencing-based assessment of soil pH as a predictor
of soil bacterial community structure at the continental scale. *Applied and Environmental Microbiology* **75**,
5111–5120 (2009).
- **38.** Blossfeld, S., Perriguet, J., Sterckeman, T., Morel, J.-L. & Lösch, R. Rhizosphere pH dynamics in trace-metal-
contaminated soils, monitored with planar pH optodes. *Plant and soil* **330**, 173–184 (2010).
- **39.** Amman, D. *Ion-selective micro-electrodes* (Springer Berlin, 1988).
- **40.** Maier, S. *et al.* Photoautotrophic organisms control microbial abundance, diversity, and physiology in different
types of biological soil crusts. *The ISME journal* **12**, 1032–1046 (2018).
- **41.** Or, D., Phutane, S. & Dechesne, A. Extracellular polymeric substances affecting pore-scale hydrologic
conditions for bacterial activity in unsaturated soils. *Vadose Zone Journal* **6**, 298–305 (2007).
- **42.** Rubol, S. *et al.* Connecting bacterial colonization to physical and biochemical changes in a sand box infiltration
experiment. *Journal of hydrology* **517**, 317–327 (2014).
- **43.** Adessi, A., de Carvalho, R. C., De Philippis, R., Branquinho, C. & da Silva, J. M. Microbial extracellular
polymeric substances improve water retention in dryland biological soil crusts. *Soil Biology and Biochemistry*
**116**, 67–69 (2018).
- **44.** Rubol, S., Manzoni, S., Bellin, A. & Porporato, A. Modeling soil moisture and oxygen effects on soil
biogeochemical cycles including dissimilatory nitrate reduction to ammonium (dnra). *Advances in water*
*resources* **62**, 106–124 (2013).
- **45.** Wu, D. *et al.* Soil N₂O emissions at high moisture content are driven by microbial nitrate reduction to nitrite:
tackling the N₂O puzzle. *The ISME journal* **1** (2019).
- **46.** Silver, W. L., Lugo, A. & Keller, M. Soil oxygen availability and biogeochemistry along rainfall and topographic
gradients in upland wet tropical forest soils. *Biogeochemistry* **44**, 301–328 (1999).
- **47.** Ebrahimi, A. & Or, D. Hydration and diffusion processes shape microbial community organization and function
in model soil aggregates. *Water Resources Research* (2015).
**48.** Caranto, J. D. & Lancaster, K. M. Nitric oxide is an obligate bacterial nitrification intermediate produced by
hydroxylamine oxidoreductase. *Proceedings of the National Academy of Sciences* **114**, 8217–8222 (2017).
**Data availability**
Source data for figures are provided with the paper. Other relevant data are available on request from the corresponding
author (MK).
**Code availability**
The MATLAB codes of the desert biocrust model (the DBM) are available on a GitHub repository at:
<http://github.com/minsughim/DBM-for-drying-soils>
**Acknowledgements**
MK thanks Daniel Breitenstein (ETHZ) for technical assistance in the lab, Daniel Baumann for IT support, Minsung
Kim for the illustration, and Samuel Bickel (ETHZ) for constructive discussion. Prof. Dr. Michael Sander (ETHZ)
kindly provided sterilised quartz sand and the rotameter for the measurements. Kurt Barmettler (ETHZ, Soil
Chemistry group, Prof. Dr. Ruben Kretzschmar) also kindly provided a multimeter for microelectrodes. Authors
acknowledge Dr. Bettina Weber (Max Planck Institute for chemistry and University of Graz) and her coworkers for
helpful comments and their data of HONO emissions from drying biocrusts. MK and DO would like to thank the
editor and the reviewers for their many insightful comments and suggestions that helped to improve the manuscript.
This work was supported by European Research Council (ERC) Advanced Grant (320499-SoilLife).
**Author contributions statement**
DO and MK conceived the research. MK wrote the code of the DBM and performed experiments. MK and DO
carried out the analysis of results and co-wrote the paper.
**Additional information**
Supplementary information is available for this paper.
**Competing financial interests**
The authors declare no competing interests.

Figure 1. A schematic of HONO and NH_3 emissions due to biotic and abiotic processes in aqueous films on grain surfaces of unsaturated soil. Nitrification performed by ammonia oxidising bacteria (AOB) and nitrite oxidising bacteria (NOB) increases or reduces affects the gas-gaseous emissions of NH_3 and HONO directly by altering the concentrations of their protonated forms within thin water film aqueous films. Increase An increase in concentration during a course of desiccation causes outgassing and precipitation of these compounds depending on the their solubility. Their partitioning and chemical speciation are determined by the partial pressure-Henry's law and acid-base equilibria. The product of aerobic nitrification, nitrate (the mixing-ratio NO_3^-) of the compound in soil-air, can accumulate locally and reach high concentrations that result in HONO emission hotspots with local film-pH, acidity. This localised and temperature highly dynamic process cannot be captured by averaged soil pH of saturated soils under static conditions.

Figure 2. Model predictions of changes in abiotic conditions during drying of soils (a) A schematics of changes in aqueous phase configurations in soils during drying. (b) A typical model calculation of water content (black solid line) and effective water film thickness (black dashed line) as a function of matric potential (blue-yellow gradient represents relative wetness). (c) A comparison of characteristic time scales for physico-chemical processes relevant for local pH determination in aqueous films for a range of hydration conditions (expressed as matric potential).

Figure 3. Laboratory measurements of pH dynamics under two wet-dry cycles monitored using a planar pH optode (pH sensor SF-HP5-OIW, PreSens GmbH, Rosensburg, Germany) and a pH microelectrode (PH-200C, Unisense, Aarhus, Denmark). (a) An illustration of the measurement cuvette and experimental setup. The optode imaging sensor was mounted at the bottom of the glass cuvette and the microelectrode was installed from the top, upright. A small glass cuvette (20 mm × 20 mm × 20 mm) filled with an agar block saturated with phosphate buffer saline (PBS pH = 6.1, 0.1M) (left) or wet quartz sand (right) while monitoring pH variations during drying. Sample desiccation was controlled by airflow rate (relative humidity 20%) into the cuvette and hydration status of the sample was monitored simultaneously by weighing the entire sample. (b) pH changes in drying agar monitored with the optode (red-magenta circles) and the microelectrode (green-turquoise circles). (c) The amount of water in the cube was measured in weight and the value was translated to equivalent water depth of the agar cube (maximal value was 4 mm). (d) pH changes during drying of wet quartz sand monitored with the optode (purple squares) and the microelectrode (orange squares). (e) variations in the hydration status of the sand expressed as gravimetric water contents (weight of water/weight of wet sand [g/g]).

Figure 4. Direct measurement of pH localisation and dynamics during desiccation of quartz sand of different textures under different atmospheric \$\text{pCO}_2\$ levels. (a) A top view of gamma-ray sterilised quartz sand with fine (0.08-0.2 mm, red box in the inset) and coarse (0.7-3 mm, yellow box in the inset) domains; optode measured pH values for different regions are shown by symbols with error bars; the dynamics of spatial pH maps during pH transition are given as inset figures at 20 min intervals (the scale bar indicates 5 mm) (b) The saturation dynamics during evaporation defined as the amount of water in the sample relative to the amount of deionised water applied for saturating the sample). (c) The variations in spatially averaged pH of the same sterilised quartz during drying for different levels of pCO_2 in the measurement cuvette. (d) Saturation dynamics during desiccation for experiments conducted under different pCO_2 levels.

Figure 5. Dynamic processes during biocrust desiccation as predicted by the desert biocrusts model (DBM). The results were obtained from 8 different simulations with identical boundary conditions ~~using numerical bioerusts~~. (a) Changes in saturation and increase in gas percolation during 24 h drying ~~(the insets schematically depict aqueous phase configurations during drying)~~. (b) Simulated production and consumption of NO₂⁻ by microorganisms; NO₂⁻ consumption by NOB (light blue) exceeds the production by AOB (dark blue). Solid lines are the averaged values and shaded areas indicate 1 ~~std for SD~~ of all simulations. (c) Mean local pH (red line) where spatial heterogeneity of local pH spans a wide range of pH values (shaded area indicates from ~~the~~ minimum to the maximum). (d) The dynamics of HONO (green) and NH₃ (purple) emissions from the model biocrust with positive and negative flux values indicate emission or uptake ~~from by~~ the domain, respectively. (e) Simulated variations in inorganic nitrogen compounds with NO₃⁻ (red), NH₄⁺ (blue), and NO₂⁻ (green) during drying (values are given in ppm with the unit ~~g.g_{soil}⁻¹~~). (f) Simulated local concentrations of NO₃⁻ in the aqueous phase plotted against local pH at 4 hours intervals (t = 0, 4, ..., 24) during drying (the colour bar corresponds to ~~mean the~~ matric potential, ~~ψ_m~~, and the values are ~~taken from a typical simulation of the simulations~~). (g) The relationship between local aqueous film pH as a function of hydration state (expressed as matric potential) and the amount of NO₃⁻ ~~in ppm~~.

Figure 6. HONO and NH₃ gaseous emissions during bioerust drying as functions of time and soil hydration conditions. Simulations of different conditions in drying patterns (Solid lines: slow drying, Dashed lines: fast drying), and atmospheric NH₃ levels (low: 5 ppb, high: 20 ppb) are presented denoted as slow/high drying with low/high NH₃ level, S-L (green), S-H (blue), F-L (orange), and F-H (red), respectively. The simulation results of (a) HONO and (b) NH₃ emissions are plotted during (c) 24 hours of drying at two rates. (d) measured and simulated emissions with measurements from several bioerusts, bioerust 1 (light crust Hydration conditions are expressed in South Africa, ²¹), bioerust 2 (dark crust in South Africa, ²¹), bioerust 3 (cyanobacteria-dominated crust in South Africa, ⁴⁰), bioerust 4 (light crust in Cyprus, ²²), and bioerust 5 (dark crust in Cyprus, ²²). (e) Simulated emissions percent of NH₃ from the same drying bioerusts are plotted (no data for comparison) water holding capacity.

Figure 7. HONO gaseous emissions during biocrust drying as a function of soil hydration conditions (expressed in percent of water holding capacity). Typical simulations of different conditions in drying patterns, and atmospheric NH_3 levels (low: 5 ppb, high: 20 ppb) are denoted as slow/high drying with low/high NH_3 level, S-L (green), S-H (blue), F-L (orange), and F-H (red), respectively. The length of each box indicates ± 1 SD and each stick ranges the minimum and maximum emission of HONO. Colour gradients indicate the averaged spatial variance of local pH values across simulations ($n=8$). (a) Simulated HONO emission with fast drying under high NH_3 input was comparable with measurements from cyanobacteria-dominated crust in South Africa⁴⁰. (b) Simulated HONO emission with slow drying under low NH_3 input was comparable with measurements from light crust in Cyprus²².

Reviewers' comments:

Reviewer #1 (Remarks to the Author):

The authors have provided a much-improved manuscript and responded to my concern that the manuscript was too narrowly framed in the earlier version, resulting in limited appeal to a broader audience. With that said, the transition to biocrusts in paragraph two is a bit awkward. The statement that, "A prominent example of tightly coupled biotic and abiotic processes is found in desert environments of arid or semi-arid regions" is not a strong lead sentence for that paragraph. I think a better approach would be to reference the previous work on the use of biocrusts as a model microbial system to address questions in community, landscape and ecosystem ecology (Bowker et al. 2014, Maestre et al. 2016).

In the introduction, I would also take more care in referencing the N budgets in desert soils publications and understand the uncertainty in these global estimates. For example, the statement "Desert soils are known to have low soil N accretion rates, with only 10% of fixed N being retained" is actually not true. In Peterjohn and Schlesinger (1991) they state that given the "limitations of existing data, a regional approximation of nitrogen fixation would be unreliable. Therefore, nitrogen inputs due to fixation will not be included in our calculation of the lower limit for nitrogen loss in desert ecosystems."

I would state this even more strongly that we should be VERY wary of regional and global estimates of N fixation especially when it relies on converting ARA data to actual N fixed for biocrust communities due to issues with the method. Although researchers keep publishing global estimates of N fixation in high profile journals, it's clear that the acetylene reduction assay method (ARA) on which most of these are based are highly uncertain, especially for biocrust communities. In the publication that was cited (Rodriguez-Caballero et al. 2018) the authors acknowledge the difficulty in converting ARA to actual N fixed with this statement:

"N fixation values of biocrust communities have mostly been determined using Acetylene reduction assays (ARA), but only rarely conversion ratios of ethylene to N₂ have been determined (see Barger et al. 2016). As these have been shown to range between 0.022 and 3.49, it would have been necessary to determine them on a regular basis or to utilize ¹⁵N₂ for determination of N fixation rates. However, these regular checks have been conducted only infrequently and to our knowledge only two studies on biocrust N fixation rates used ¹⁵N₂ until now. Thus, these limitations in N fixation rates have to be kept in mind."

Following this, I would suggest that the global estimates of N fixation in drylands not be so prominently featured in the introduction, since the estimate are highly uncertain.

Bowker et al. 2014. Biological soil crusts (biocrusts) as a model system in community, landscape and ecosystem ecology. *Biodiversity and Conservation*. 23(7): 1619-1637.

Maestre, FT et al. Biological Soil Crusts: An Organizing Principle in Drylands(2016):Biological Soil Crusts as a Model System in Ecology. *Ecological Studies Volume 226 Chapter 20*.

Reviewer #2 (Remarks to the Author):

I enjoyed reading the revised version of the manuscript 'Microscale pH variations during drying of soils and desert biocrusts affect HONO and NH₃'. The authors did an excellent job in implementing the reviewers' comments.

The revised version is appealing to a broad audience of soil scientists, it is focusing on both soils and biocrusts and includes relevant considerations on the N-partitioning.

I particularly enjoyed the new figure 1, which clearly illustrates the main message of the paper.

In my opinion, this paper represents a nice contribution to the journal.

I only have a minor suggestion:

Line 32 page 10

For the seek of clarity, the authors should specify the limits of the proposed model in term of N losses. For instance, N₂O may not be always negligible in aerobic soils. Several processes such as nitrifier denitrification, aerobic denitrification and so on may be a significant source of N₂O in upland soils. Nitrification itself may produce N₂O under specific conditions. In addition, other pathways such as nitrate ammonification (the conversion of ammonium to nitrate) may affect the magnitude of the N-fluxes described in this work. (e.g., see [33] of the revised version). While it is not realistic to incorporate all these processes in a model (we only have limited knowledge of most of the mechanisms regulating these pathways), it should be mentioned that they exist and may affect the N-loss balance. In addition, several experimental works showed that even in aerobic soils, anoxic and anaerobic micro-niches may exist. See for instance:

-Production of NO and N₂O by soil nitrifying bacteria, Nature 1981

-Anoxic microsites in upland soils dominantly controlled by clay content, Soil Biology and Biochemistry, Volume 118, 2018, Pages 42-50

-2D visualization captures the local heterogeneity of oxidative metabolism across soils from diverse land-use, Science of the total environment 2016

-Nitrous oxide production by nitrification and denitrification in soil aggregates as affected by O₂ concentration, Soil Biology and Biogeochemistry 2004

Response to Reviewers' Comments: NCOMMS-18-28408A

Microscale pH variations during drying of soils and desert biocrusts affect HONO and NH₃ emissions

Minsu Kim^{1,2*} and Dani Or¹

¹ Soil and Terrestrial Environmental Physics (STEP),
Department of Environmental Systems Sciences (USYS), ETH Zürich,
8092 Zürich, Switzerland

² Laboratory for Air Pollution/Environmental Technology,
Empa (Swiss Federal Laboratories for Materials Science and Technology),
8600 Dübendorf, Switzerland

Correspondence to: Minsu Kim (minsukim@empa.ch)

We sincerely thank the reviewers and the editor for the positive comments and suggestions that improved the manuscript. In the following, we provide a point-by-point response to all comments.

Reviewers' comments:

Reviewer #1 (Remarks to the Author):

The authors have provided a much-improved manuscript and responded to my concern that the manuscript was too narrowly framed in the earlier version, resulting in limited appeal to a broader audience. With that said, the transition to biocrusts in paragraph two is a bit awkward. The statement that, "A prominent example of tightly coupled biotic and abiotic processes is found in desert environments of arid or semi-arid regions" is not a strong lead sentence for that paragraph. I think a better approach would be to reference the previous work on the use of biocrusts as a model microbial system to address questions in community, landscape and ecosystem ecology (Bowker et al. 2014, Maestre et al. 2016).

We thank the reviewer for this comment. We have changed the transition from the first to the second paragraph following the suggestion (Line 15, page3)

In the introduction, I would also take more care in referencing the N budgets in desert soils publications and understand the uncertainty in these global estimates. For example, the statement “Desert soils are known to have low soil N accretion rates, with only 10% of fixed N being retained” is actually not true. In Peterjohn and Schlesinger (1991) they state that given the “limitations of existing data, a regional approximation of nitrogen fixation would be unreliable. Therefore, nitrogen inputs due to fixation will not be included in our calculation of the lower limit for nitrogen loss in desert ecosystems.” I would state this even more strongly that we should be VERY wary of regional and global estimates of N fixation especially when it relies on converting ARA data to actual N fixed for biocrust communities due to issues with the method. Although researchers keep publishing global estimates of N fixation in high profile journals, it’s clear that the acetylene reduction assay method (ARA) on which most of these are based are highly uncertain, especially for biocrust communities. In the publication that was cited (Rodriguez-Caballero et al. 2018) the authors acknowledge the difficulty in converting ARA to actual N fixed with this statement:

“N fixation values of biocrust communities have mostly been determined using Acetylene reduction assays (ARA), but only rarely conversion ratios of ethylene to N₂ have been determined (see Barger et al. 2016). As these have been shown to range between 0.022 and 3.49, it would have been necessary to determine them on a regular basis or to utilize 15N₂ for determination of N fixation rates. However, these regular checks have been conducted only infrequently and to our knowledge only two studies on biocrust N fixation rates used 15N₂ until now. Thus, these limitations in N fixation rates have to be kept in mind.”

Following this, I would suggest that the global estimates of N fixation in drylands not be so prominently featured in the introduction, since the estimate are highly uncertain.

We removed the global estimates of N fixation from the introduction and refined the paragraph focusing on our objective to identify N loss mechanisms (Line 15-25, page3):

“The quantification of interactions between biotic and abiotic processes in soil remains a challenge, yet progress has been made in certain microbial systems, such as soil aggregates⁵ and biological soil crusts⁶, that help disentangle their role in ecosystem functioning. Biological soil crusts (hereafter biocrusts) have been suggested as a model microbial system to study microbial interaction at the community level within a well-defined domain (crust) under various abiotic conditions^{7,8}. Biocrusts develop in cold and warm deserts environments. Despite water limitations, these thin crusts host dense microbial communities and contribute significantly to biological N_r exchanges with the atmosphere^{9,10}. Considering that biocrusts are active only when wet, the partitioning and fate of imported N_r during wetting events are of particular importance for their surrounding environments. N can be a limiting nutrient for desert ecosystems owing to relatively high loss of N_r as gaseous emission¹¹. However, the

picture of the nitrogen balance in biocrusts is more complicated due to strong effects of surface wetness, temperature, and community composition on N_r dynamics¹²."

Bowker et al. 2014. Biological soil crusts (biocrusts) as a model system in community, landscape and ecosystem ecology. Biodiversity and Conservation. 23(7): 1619-1637.

Maestre, FT et al. Biological Soil Crusts: An Organizing Principle in Drylands(2016):Biological Soil Crusts as a Model System in Ecology. Ecological Studies Volume 226 Chapter 20.

We added these two references in the main text.

Reviewer #2 (Remarks to the Author):

I enjoyed reading the revised version of the manuscript 'Microscale pH variations during drying of soils and desert biocrusts affect HONO and NH₃'. The authors did an excellent job in implementing the reviewers' comments. The revised version is appealing to a broad audience of soil scientists, it is focusing on both soils and biocrusts and includes relevant considerations on the N-partitioning. particularly enjoyed the new figure 1, which clearly illustrates the main message of the paper.

In my opinion, this paper represents a nice contribution to the journal.

We thank the reviewer for the positive comments. We were also happy to read that the revised manuscript is enjoyable and is possibly appealing to a broad audience.

I only have a minor suggestion:

Line 32 page 10 For the seek of clarity, the authors should specify the limits of the proposed model in term of N losses. For instance, N₂O may not be always negligible in aerobic soils. Several processes such as nitrifier denitrification, aerobic denitrification and so on may be a significant source of N₂O in upland soils. Nitrification itself may produce N₂O under specific conditions. In addition, other pathways such as nitrate ammonification (the conversion of ammonium to nitrate) may affect the magnitude of the N-fluxes described in this work. (e.g., see [33] of the revised version). While it is not realistic to incorporate all these processes in a model (we only have limited knowledge of most of the mechanisms regulating these pathways), it should be mentioned that they exist and may affect the N-loss balance. In addition, several experimental works showed that even in aerobic soils, anoxic and anaerobic micro-niches may exist. See for instance:

-Production of NO and N₂O by soil nitrifying bacteria, Nature 1981

-Anoxic microsites in upland soils dominantly controlled by clay content, Soil Biology and Biochemistry, Volume 118, 2018, Pages 42-50

-2D visualization captures the local heterogeneity of oxidative metabolism across soils from diverse land-use, Science of the total environment 2016

-Nitrous oxide production by nitrification and denitrification in soil aggregates as affected by O₂ concentration, Soil Biology and Biogeochemistry 2004

We agree on stating the limitation of the model in terms of N gases emission and different pathways that are not included in the model. We added this in discussion together with the suggested references in the main text (Line 2-12, page 11).

" Here we focused primarily on aerobic processes, such as nitrification, since desert and near-surface soils are often dry and mostly aerated (shown in Fig.5 of Kim and Or, 2017⁶). Furthermore, we also observed that, in our simulations, the heterotrophic activity of

denitrifiers was inhibited due to the limited extent of anoxic regions and the absence of carbon sources, notwithstanding their presence in the model. In other soil systems with sufficient carbon sources, the presence of anoxic or anaerobic microsites is an important factor even near the soil surface especially when oxygen consumption by aerobic organisms and shallow roots may exceed its diffusion rates into the soil⁴². The conditions within soil aggregates and in fine textured soils with appreciable EPS promote the formation and persistence of anoxic microsites, that, in turn, may affect N-losses following wetting due to anaerobic production of N₂O for instance^{5,43,44}. We should mention that the model does not include other pathways, such as nitrifier denitrification⁴⁵ or nitrate ammonification⁴⁶ that produce N₂O or NO and could affect the estimation of N gaseous effluxes reported in this study. "